# Rapid human oogonia-like cell specification via transcription factor-directed differentiation

Merrick Pierson Smela[1,2], Christian C Kramme [1,2✉], Patrick R J Fortuna[1,2], Bennett Wolf[1,2], Shrey Goel[3], Jessica Adams[1,2], Carl Ma [1,2], Sergiy Velychko [1,2], Ursula Widocki [4], Venkata Srikar Kavirayuni[3], Tianlai Chen[3], Sophia Vincoff[3], Edward Dong[1,2], Richie E Kohman [1,2], Mutsumi Kobayashi [5], Toshi Shioda [6], George M Church[1,2] & Pranam Chatterjee[3,7,8]

## Abstract

The generation of germline cells from human induced pluripotent stem cells (hiPSCs) represents a milestone toward in vitro gametogenesis. Methods to recapitulate germline development beyond primordial germ cells in vitro have relied on long-term cell culture, such as 3-dimensional organoid co-culture for ~four months. Using a pipeline with highly parallelized screening, this study identifies combinations of TFs that directly and rapidly convert hiPSCs to induced oogonia-like cells (iOLCs). We demonstrate that co-expression of five TFs – namely, *ZNF281, LHX8, SOHLH1, ZGLP1,* and *ANHX*, induces high efficiency DDX4-positive iOLCs in only four days in a feeder-free monolayer culture condition. We also show improved production of human primordial germ cell-like cells (hPGCLCs) from hiPSCs by expression of *DLX5, HHEX,* and *FIGLA*. We characterize these TF-based iOLCs and hPGCLCs via gene and protein expression analyses and demonstrate their similarity to in vivo and in vitro-derived oogonia and primordial germ cells. Together, these results identify new regulatory factors that enhance human germ cell specification in vitro, and further establish unique computational and experimental tools for human in vitro oogenesis research.

**Keywords** Oogonia; Primordial Germ Cells; Differentiation; Induced Pluripotent Stem Cells; Transcription Factors
**Subject Categories** Chromatin, Transcription & Genomics; Methods & Resources; Stem Cells & Regenerative Medicine

## Introduction

In the human germline, oogonia are specified from primordial germ cells (PGCs) that undergo sequential differentiation through sex-specific trajectories to form female gametes (Kobayashi et al, 2017). In vitro gametogenesis (IVG) is a powerful technique for the recapitulation of germline development, often utilizing growth factor and co-culture-directed differentiation of pluripotent stem cells to sequentially specify discrete germline cell types (Saitou and Hayashi, 2021). Germ cells are challenging to obtain for research in humans due to inherent ethical and technical limitations in their in vivo isolation and utilization, since these critical intermediates span embryonic, fetal, and adult development stages (Nagaoka et al, 2021). With the rate of infertility on the rise globally, new cell models of human reproductive development are needed to accelerate research into sex cell formation and genetic regulation, investigate root causes of infertility, and develop novel assisted reproductive technologies (NSFG - listing I - key Statistics from the National Survey of family growth, 2019).

Methods for generating primordial germ cell-like cells (hPGCLCs), which correspond to pre-migratory PGCs in vivo, from stem cells are abundant and utilize a variety of both three-dimensional and two-dimensional methodologies (Saitou and Hayashi, 2021; Gell et al, 2020; Irie et al, 2015; Kobayashi et al, 2017, 2022; Mitsunaga et al, 2017; Sebastiano et al, 2021; Kojima et al, 2021, 2017; Yokobayashi et al, 2017; Chen et al, 2017). Recently, human DDX4+ oogonia were specified from pluripotent stem cells through a ~4-month chimeric culture model in xenogeneic reconstituted ovaries (xrOvaries) (Yamashiro et al, 2018). This remarkable work demonstrated that advanced human germ cell types can be differentiated from stem cells in vitro, a key step toward IVG. However, this impressive method remains challenging to utilize, particularly in screening paradigms that require scaled, parallelizable cell type differentiation, due to the extensive culture periods in 3D chimeric aggregates and air-liquid interface culture, which require fetal mouse gonad dissection combined with human primordial germ cell-like cell specification. To date, no other method yet exists for the direct generation of oogonia from stem cells in humans. A related method of generating DDX4+ germ cells was recently reported using a specialized mouse fibroblast line (m220) instead of fetal ovarian somatic cells to support multi-month hPGCLC culture, but these germ cells did not

[1]Wyss Institute, Harvard Medical School, Boston, MA, USA. [2]Department of Genetics, Harvard Medical School, Boston, MA, USA. [3]Department of Biomedical Engineering, Duke University, Durham, NC, USA. [4]Broad Institute of Harvard and MIT, Cambridge, MA, USA. [5]Department of Obstetrics and Gynaecology, School of Medicine, Juntendo University, Tokyo, Japan. [6]Massachusetts General Hospital Krantz Family Center for Cancer Research, Harvard Medical School, Boston, MA, USA. [7]Department of Computer Science, Duke University, Durham, NC, USA. [8]Department of Biostatistics and Bioinformatics, Duke University, Durham, NC, USA. ✉E-mail: christian@gametogen.com

silence pluripotency gene expression as observed in xrOvary or in vivo oogonia (Murase et al, 2024).

In mouse models, several studies have demonstrated complete female IVG, yielding developmentally competent metaphase II (MII) oocytes from stem cells (Hikabe et al, 2016; Yoshino et al, 2021). Recent mouse studies have also demonstrated the utility of transcription factor (TF) based methods for driving and enhancing germ cell development (Nagaoka et al, 2020; Hamazaki et al, 2021). These TF-based methods benefit from their ease of use, rapid developmental timescales, and high efficiency. TF-driven specification of hPGCLCs has also been demonstrated, with seminal work identifying that the TFs *SOX17*, *TFAP2C*, and *PRDM1* and the GATA members *GATA3/2* comprise an essential network for hPGCLC specification (Kojima et al, 2017, 2021; Irie et al, 2015; Kobayashi et al, 2017; Sasaki et al, 2015). However, apart from this upstream TF network for hPGCLC specification, the roles of downstream TFs in driving later stages of human germline development have not been closely examined, likely due to a combination of challenging cell culture techniques, difficulty in establishing screening methods, and lack of compelling TF target prediction tools. In particular, no TF induction-based methods have yet been demonstrated for deriving more advanced human germ cell types such as oogonia, spermatogonia, or oocytes from stem cells.

In this study, we identify novel combinations of TFs that can direct the differentiation of human induced pluripotent stem cells (hiPSCs) to distinct germ cell types, including DDX4+ induced oogonia-like cells (iOLCs) as well as hPGCLCs. Based on previously developed graph theory algorithms, we infer TFs with highly connected roles in the later stages of female gamete development (Kramme et al, 2021a). Using hiPSC lines engineered with reporter alleles for different stages of germline development, we establish a high-throughput screening system to identify TFs that induce reporter-positive human germ cell types. We show that individual overexpression of the TFs *DLX5*, *HHEX*, and *FIGLA* drives potent enhancement of canonical hPGCLC formation. We likewise identify *LHX8*, *SOHLH1*, and *ZNF281* as TFs whose combinatorial overexpression directly generates DDX4+ iOLC formation in four days in monolayer culture. We furthermore demonstrate a method for feeder-free post-isolation culture of the DDX4+ iOLCs that utilizes recently described long-term hPGCLC expansion and maintenance methods (Kobayashi et al, 2022). Finally, we describe an improved protocol for DDX4+ iOLC production that uses DNA methyltransferase inhibition and overexpression of the additional TFs *ANHX* and *ZGLP1* to achieve higher iOLC yields and an epigenetic state that more closely resembles in vivo oogonia. Using immunofluorescence, transcriptomic, and epigenomic assays, we demonstrate that these TF-based PGCLCs and iOLCs closely resemble in vivo and in vitro human germ cells. Together, these results demonstrate the potential of TF-based methods for generating human germ cell types, and provide a rapid and accessible system for producing oogonia-like cells in vitro.

# Results

## In silico identification of transcription factors that are central regulators of human oogenesis

Previous studies to identify regulators of mouse oogenesis examined differential gene expression of the primordial-to-primary follicle transition, identifying TFs that regulated mouse oocyte maturation

and were functionally capable of driving oocyte-like cell formation from stem cells (Hamazaki et al, 2021). Similarly, to identify candidate TFs that govern the dynamic gene regulatory networks (GRNs) of human oogenesis, we curated publicly available single-cell RNA-sequencing (scRNA-seq) data of ovarian folliculogenesis and normalized the gene expression of germ cells at various stages of follicle development, including primordial, primary, secondary, antral, and preovulatory follicles (Zhang et al, 2018b; Li et al, 2017; Yamashiro et al, 2018; Kobayashi et al, 2022). After constructing this pseudo-time series transcriptomic matrix, we employed unsupervised, regression-based GRN inference via regularized stochastic gradient boosting to generate a representative, directed graph of oocyte state during folliculogenesis, where nodes represent genes and edge weights correlate to the regulatory effect between TFs and their downstream targets (Moerman et al, 2019). After conducting standard pruning of the graph for low-weight, spurious edges, we applied the adapted PageRank algorithm from our recently published STAMPScreen pipeline to prioritize the 36 most critical, "central" TFs within the GRN (Fig. 1A; Dataset EV1) (Kramme et al, 2021a). Our ranked list includes both previously-identified regulators of mammalian oocyte transcriptional networks and germ cell development, such as *NOBOX*, *LHX8*, and *OTX2*, as well as potential novel regulators, including *ZNF281*, *SOX13*, and *DLX5* (Zhang et al, 2018a; Hamazaki et al, 2021). We additionally include 10 TFs that ranked slightly lower on our predictive list (which only leveraged late-stage oogenesis data), but are known to perform important roles in earlier specific stages of germ cell development such as PGC formation, or that have been identified in similar mouse studies (Hamazaki et al, 2021; Kojima et al, 2021, 2017; Nagaoka et al, 2020). The list of the 36 TFs and ten control TFs is presented in Dataset EV2 with the isoform chosen for screening, if it was predicted or selected as a control, and any known involvement in germ cell development. Associated GO term analysis of the 46 factors is shown in Fig. 1B, with reproduction and oogenesis terms highlighted as well as select general GO terms. We furthermore assessed the expression of all 46 TFs in single-cell RNA-sequencing (scRNA-Seq) datasets from the recently described human fetal gonad atlas, which is shown in Appendix Fig. S1 (Garcia-Alonso et al, 2022). As can be seen, each of the 46 TFs is expressed in the fetal gonad germ cells in both male and female development. Overall, these results motivated us to screen prioritized candidates in hiPSCs to resolve the human oocyte transcriptional network experimentally.

## Development of a TF-overexpression screening platform in hiPSCs for germ cell-reporter formation

We constructed a DDX4-T2A-tdTomato; NANOS3-T2A-mVenus dual reporter female hiPSC line (D4TN3V) using CRISPR-Cas9-mediated homology-directed repair (Appendix Fig. S2A) and confirmed proper reporter insertion by whole genome sequencing (Appendix Fig. S2B). The cell line was confirmed karyotypically normal and pluripotent using the Karyostat and Pluritest assays (Appendix Fig. S2C,D). Using flow cytometry after CRISPR-activation induction of the DDX4 reporter and floating aggregate hPGCLC induction, we confirmed that the reporter lines induce expression of the fluorescent markers upon induction of the target genes, and for the NANOS3 reporter, overlaps with known hPGCLC markers EpCAM/ITGA6 (Appendix Fig. S2E,F). For screening purposes, a 4-day monolayer protocol, differentiating cells to epiblast-like intermediates followed by BMP4 induction,

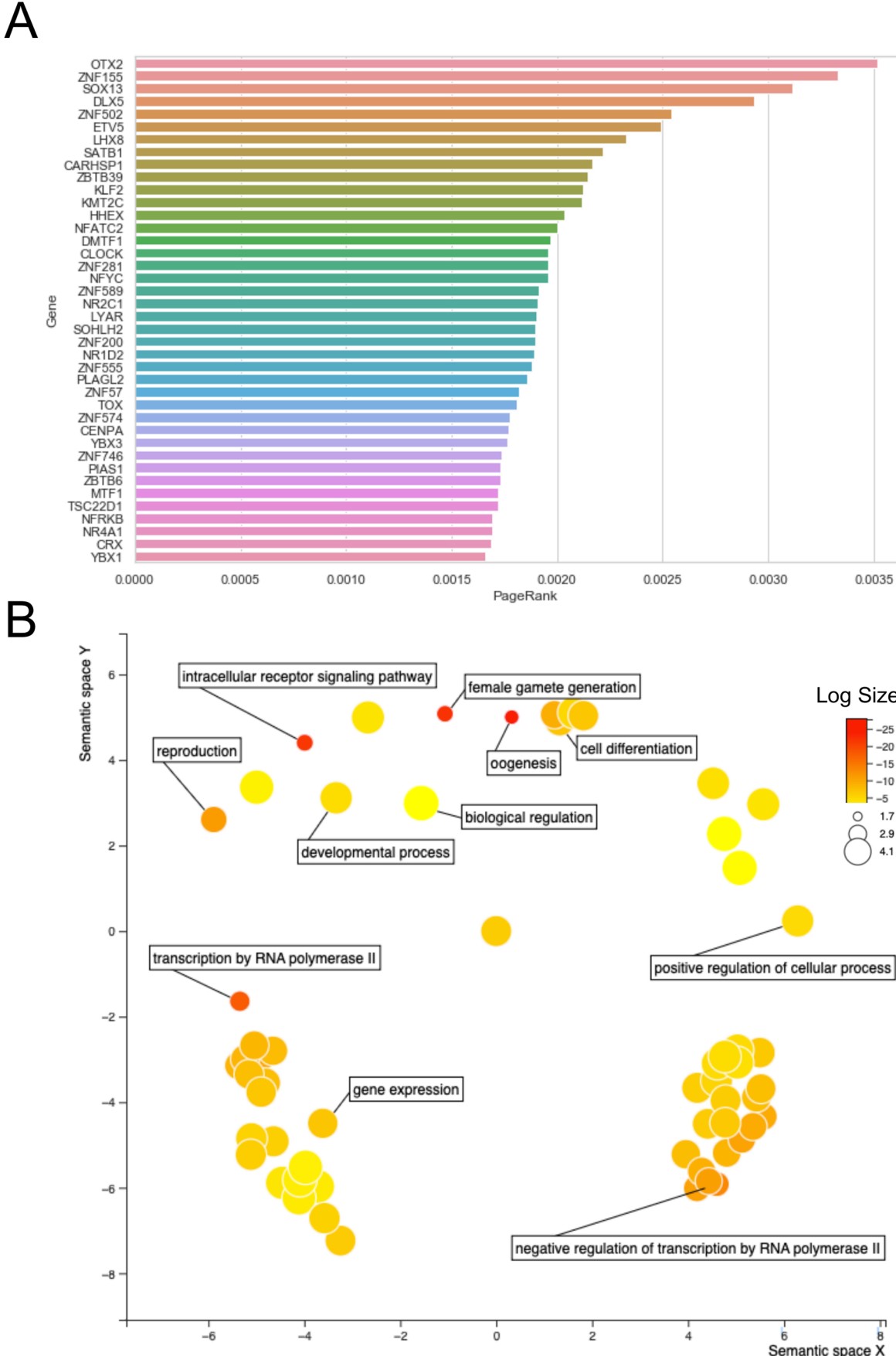

◀ **Figure 1. In silico identification of TFs with putative germ cell formation regulatory functions.**

(A) Inferred gene regulatory network generated from folliculogenesis dataset of Zhang et al, 2018b. with a prioritized TF screening list, ordered by PageRank score imputed from the gene regulatory network and centrality algorithm. (B) All 46 TFs utilized in the study were analyzed using the GO term tool g:profiler, and significant GO terms were plotted in Revigo visualizer highlighted terms related to significantly enriched pathways for germline development and transcription factor function. Significantly enriched terms are determined by Fisher's one-tailed test, also known as cumulative hypergeometric probability, as the *p* value measuring the randomness of the intersection between the query and the ontology term, with $p < 0.05$ considered significant. Color and dot size reflect the log size of the *p* adjusted value.

was adapted from monolayer hPGCLC generation methods described previously (Fig. 2A) (Sebastiano et al, 2021). We chose monolayer methods for screening due to the ease and scalability of use compared to floating aggregate methods (Jinek et al, 2013; Irie et al, 2015; Yamashiro et al, 2018). We generated doxycycline-inducible vectors expressing a full-length cDNA for each of the 46 TFs using a highly optimized piggyBac backbone vector for tunable expression in hiPSCs from our STAMPScreen pipeline (Kramme et al, 2021b); (Yusa et al, 2011). We generated 46 individual D4TN3V hiPSC lines harboring integrations of each TF individually through super piggyBac transposase-mediated insertion (Appendix Fig. S3A). Polyclonal pools for each TF were utilized for screening purposes. Through induction of each individual line via doxycycline followed by 3' UTR barcode capture in RNA-seq, we demonstrated expression of the intended TF and its associated barcode in all 46 lines utilized in the study, confirming each line's genotype and functional gene expression of each vector (Appendix Fig. S3B). We additionally utilized the *DLX5* overexpression cell line and validated that TF expression in our system is highly linked to doxycycline concentration, as expected, reaching levels of ~1000-fold induction compared to the no doxycycline condition in hiPSCs determined by qPCR (Appendix Fig. S3C).

## Identification of TFs regulating reporter-positive cell formation via overexpression screening

We first assessed NANOS3+ cell yield via flow cytometry in the presence or absence of doxycycline for all 46 TFs in triplicate (Fig. EV1). We show that 23 TFs (15 computationally predicted and eight controls) drove a statistically significant increase in NANOS3+ cell yield compared to the no TF control, while the other 23 TFs demonstrated a trend towards upregulation, but not significantly. Remarkably, three TFs (*DLX5*, *HHEX*, and *FIGLA*) induced a NANOS3+ yield higher (23.6 ± 2.4% SEM, 10.5 ± 0.9% SEM, and 10.3 ± 0.7% SEM) than that of three known TF regulators of hPGCLC development: *SOX17*, *TFAP2C*, and *PRDM1* (8.6 ± 0.5% SEM, 8.5 ± 0.9% SEM, and 8.0 ± 1% SEM), respectively (Kobayashi et al, 2017; Kojima et al, 2017; Sasaki et al, 2015). To validate that this method improved canonical hPGCLC formation, we further evaluated the phenotype of the *DLX5*, *HHEX*, and *FIGLA*-induced NANOS3+ cells. Using conventional floating aggregate hPGCLC induction methods, we demonstrated that individual overexpression of *DLX5*, *HHEX*, and *FIGLA* drove a statistically significant increase in putative hPGCLC formation as assessed by the canonical surface markers EpCAM/ITGA6 in the plus doxycycline condition (11.7 ± 0.43%, 24.4 ± 0.59%, 22.9 ± 0.42%) versus minus doxycycline condition (2.1 ± 0.61%, 3.2 ± 0.12%, 3.5 ± 0.47%) ($p < 0.0001$) (Fig. EV2A) (Sasaki et al, 2015). Putative hPGCLC yield was significantly lower in the no TF control condition and did not significantly differ between the plus

doxycycline (2.6 ± 0.48%) and minus doxycycline conditions (2.2 ± 0.30%). We demonstrated through RNA-sequencing that these TF-driven NANOS3+ cells displayed canonical upregulation of *SOX17*, *PRDM1*, *TFAP2C* and downregulation of *SOX2* gene expression compared to the no doxycycline hiPSC controls (Fig. EV2B). Likewise, using immunofluorescence imaging, we demonstrated the unsorted germ cell populations in floating aggregate displayed SOX17, OCT4, and ITGA6 triple positive cells, key features of hPGCLCs, which were more abundant in the TF-overexpression floating aggregates than in the No-TF control, in line with the flow cytometry findings (Fig. EV3). Together, these results identify individual overexpression of *DLX5*, *HHEX*, and *FIGLA* improves canonical hPGCLC formation.

We next assessed the production of DDX4+ cells from hiPSCs via flow cytometry in the presence or absence of doxycycline for all 46 TFs in triplicate (Fig. 2A). DDX4 is broadly expressed throughout germ cell development, arising in gonadal hPGCs after migration to the fetal gonad and increasing in expression during oogonia/spermatogonia formation. Compared to the control, the overall percentage of DDX4+ cells was not greatly enriched by any single TF (Fig. 2B, y-axis). However, a small percentage of cells with significantly elevated DDX4-reporter expression intensity was identified in the *ZNF281*, *LHX8*, and *SOHLH1* induction conditions (Fig. 2B, x-axis). We hypothesized that combinatorial expression of these factors may increase DDX4-reporter yield and generate a more reproducible yield of highly DDX4 reporter-positive cells. To investigate this hypothesis, we generated three independent D4TN3V cell lines harboring a polyclonal integration of all three TFs, which we term DDX4 by 3 TFs (D3), and assessed the induced DDX4+ yield compared to a no TF control in biological triplicate (Fig. 2C). We find the D3 combination drives a statistically significant increase in DDX4-reporter cell yield (1.22 ± 0.27% SEM, $p = 0.0228$). Remarkably, this high DDX4+ population is obtained in just 4 days in monolayer through direct TF induction during a primordial germ cell-like cell differentiation protocol.

## An improved protocol allows the production of DDX4+ cells with higher efficiency

Although the expression of our D3 TF combination reliably produces DDX4+ cells, the efficiency is low. Therefore, we sought to identify factors which could improve this. We generated a library of barcoded PiggyBac transposon plasmids for Dox-inducible expression of 62 additional oogonia-related TFs and RNA-binding proteins, as well as 31 constitutively active signaling proteins (Martz et al, 2014). Additionally, we included a CRISPRi vector targeting *DNMT1*, since suppression of DNA methylation is known to be crucial for germ cell development (Tang et al, 2016). We integrated this library into cells which already contained Dox-inducible D3 TFs, induced oogonia-like cells, and performed

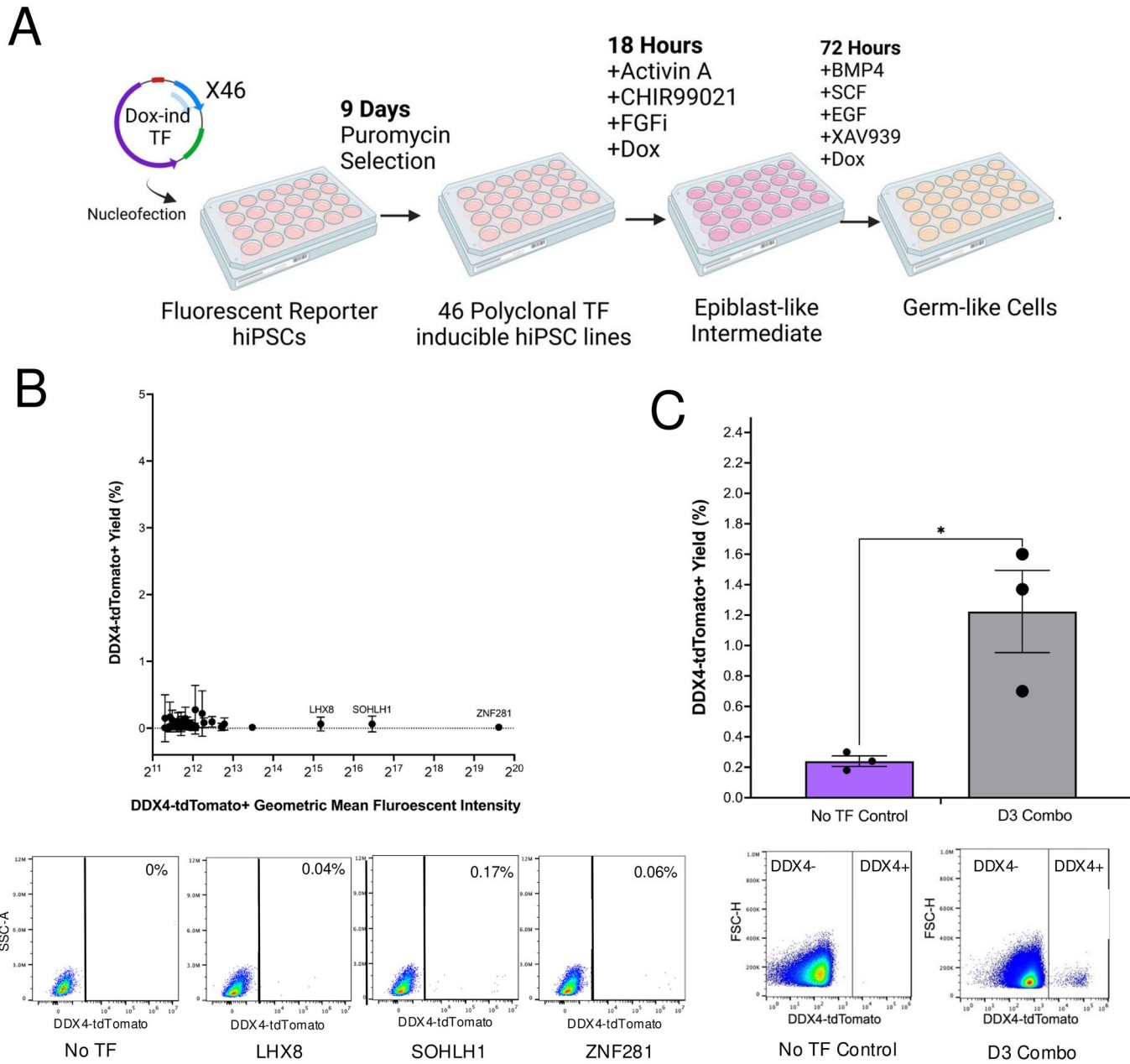

**Figure 2. Combinatorial *ZNF281*, *LHX8*, and *SOHLH1* (D3) overexpression drives rapid DDX4+ cell formation.**

(A) Schematic workflow for TF-overexpression screening, with time intervals and media supplements. (B) Flow Cytometry results of the screen for the DDX4-T2A-tdTomato reporter are visualized with percent reporter positive on the y-axis, and geometric mean fluorescence intensity of the tdTomato+ cells on the y-axis. Error bars represent the SEM of the induction yield between the triplicates in the plus dox condition. (C) Flow cytometry results are visualized for DDX4-T2A-tdTomato reporter yield (percent) for the no TF control and the D3 combo in $n = 6$ replicates (3 independent hiPSC lines, with 2 replicates per line). Individual dots represent the average for the technical replicates for each biological replicate. Data were presented as a mean ± SEM. Statistical significance was determined by a two-sided *T*-test comparison between each combination and the no TF control, with a *p* value <0.05 considered as significant. *$p = 0.0228$. Representative flow cytometry results are visualized for both the no TF control and the D3 combination. Source data are available online for this figure.

barcode enrichment screening (Fig. 3A). We observed that the top hit was *DNMT1* CRISPRi, and we selected a sub-library of 24 factors for a follow-up round of barcode enrichment screening (Fig. 3B), which we performed in both male and female DDX4 reporter hiPSCs. In this screen, we treated the cells with 5 μM GSK-3484862, a noncovalent inhibitor of DNMT1, to inhibit DNA

methylation (Azevedo Portilho et al, 2021; Haggerty et al, 2021). Additionally, we switched to a basement membrane extract overlay monolayer induction protocol which allowed higher cell seeding densities (Overeem et al, 2023). In some samples, we also tested the retinoic acid receptor agonist AM580, but we observed that this inhibited cell proliferation, and we did not include it in the final

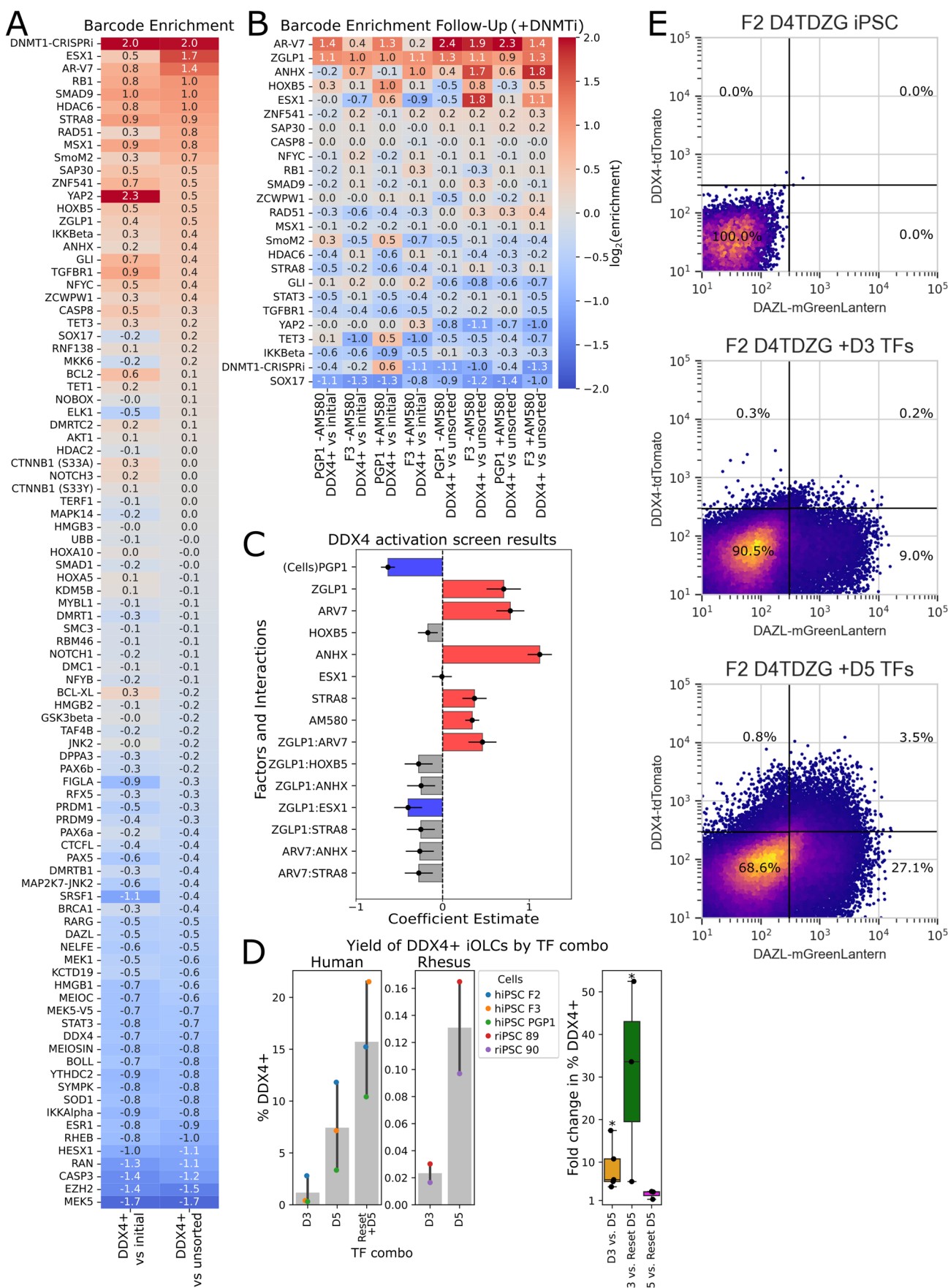

**Figure 3. Optimization of the iOLC induction protocol.**

(A) Initial barcode enrichment screen. Factors are plotted in order of log2-fold enrichment in DDX4+ cells vs. unsorted cells. (B) Followup barcode enrichment screen. A smaller set of factors were tested in an increased number of conditions. +AM580 denotes the addition of the retinoic acid receptor agonist AM580. In all conditions for this experiment, cells were treated with DNMT1 inhibitor, meaning that the effect of DNMT1 CRISPRi was masked. (C) Results of a fractional factorial screen of top factors identified in the previous experiment. Sixteen combinations were tested in two cell lines ($n = 32$). A linear model was fit to the logit-transformed percentage of DDX4+ cells. Coefficients are plotted, with error bars representing the standard errors of the coefficient estimates. (D) Comparing yield of iOLCs with D3 vs. D5 TFs in the improved protocol (BME overlay induction, +DNMT1 inhibitor). A total of $n = 3$ human and $n = 2$ rhesus monkey iPSC lines were tested. Additionally, transient naïve resetting was tested in combination with D5 TFs for $n = 3$ human iPSC lines. Error bars represent minima and maxima. Significance was calculated by two-tailed paired $t$-tests on logit-transformed data (D3 vs. D5: $n = 5$ per group, $p = 0.024$; D3 vs. reset D5: $n = 3$ per group, $p = 0.034$; D5 vs. reset D5: $n = 3$ per group, $p = 0.104$). Asterisks represent $p < 0.05$ (E) Activation of DDX4-tdTomato and DAZL-mGreenLantern reporter expression by D3 and D5 TF combinations.

protocol out of concern that it could induce premature differentiation of oogonia. After this second round of screening, we chose six promising factors and performed a fractional factorial screen, testing 16 combinations in two cell lines. These experiments identified *ANHX* and *ZGLP1* as TFs which could increase the yield of DDX4+ cells when co-expressed with the D3 TFs (Fig. 3C). We term this improved combination D5.

We validated the improved performance of the D5 TFs by comparing them with D3 TFs in three hiPSC DDX4-T2A-tdTomato reporter lines (two female, one male), and two rhesus macaque iPSC DDX4-T2A-tdTomato reporter lines (both male). Among all five lines tested, the yield of DDX4+ cells was significantly higher using D5 TFs (average 8.75-fold increase; paired $t$-test, $p = 0.02$) (Fig. 3D). We additionally tested using transient naïve resetting to improve the yield of DDX4+ cells, since this had been reported as a method for efficient germ cell induction (Alves-Lopes et al, 2023). Naïve resetting of hiPSCs with Sox2$^{A61V}$-Klf4-Myc episome, followed by DDX4+ cell induction using D5 TFs, allowed the generation of DDX4+ cells with good efficiency (Fig. 3D) (MacCarthy et al, 2024). The yield was significantly higher than for D3 TFs (average 30.5-fold increase; paired $t$-test, $p = 0.03$) but the increase in yield over D5 TFs did not reach statistical significance (average 2.5-fold increase; paired $t$-test, $p = 0.10$). Finally, we examined the co-expression of the oogonia markers DAZL and DDX4 during DDX4+ cell induction with D3 and D5 TFs, using a DDX4-T2A-tdTomato/DAZL-T2A-mGreen-Lantern (D4TDZG) dual reporter line (Pierson Smela et al, 2024). By flow cytometry, we observed expression of both markers in D3 and D5 TF conditions (Fig. 3E). However, the D5 combination had more marker-positive cells, and a greater fraction of the DDX4+ cells also co-expressed DAZL. Overall, our optimized protocol, which combines additional TFs, DNMT1 inhibition, and higher cell seeding density, enabled us to access larger numbers of DDX4 + / DAZL+ cells for use in subsequent experiments.

## TF-directed DDX4+ cells are capable of post-isolation, feeder-free culture and display an induced oogonia-like cell (iOLC) marker expression

In vivo, primordial germ cells and oogonia develop and mitotically expand, establishing the human ovarian reserve (Yamashiro et al, 2018; Wallace and Kelsey, 2010). Recent methods have been developed for expansion of hPGCLCs in vitro post-isolation (Gell et al, 2020; Murase et al, 2020; Kobayashi et al, 2022). No method yet exists for the post-isolation culture of human oogonia or oogonia-like cells in vitro after FACS isolation. We thus sought to determine if our DDX4+ cells could be cultured further post-isolation. Serendipitously, we found

that by seeding isolated DDX4+ cells onto matrigel-coated plates feeder-free in a media composition recently described for hPGCLC expansion, we were able to expand the DDX4+ cell population for at least 28 days (Fig. 4A) (Kobayashi et al, 2022). After FACS isolation, these D3 DDX4+ cells slowly divided for 28 days, retaining expression of DDX4, as can be seen by live cell fluorescent imaging of the tdTomato reporter protein (Fig. 4A). These DDX4+ retained robust expression of the DDX4-T2A-tdTomato reporter, with clear visualization of the cleaved, diffuse fluorescent protein. Since our reporter was not a fusion protein to DDX4 itself, we saw the expected pattern of fluorescent protein in both the nucleus and cytoplasm as opposed to just the cytoplasm.

Additionally, we performed immunofluorescence imaging on fixed cells at various stages, examining the oogonia markers DDX4 and DAZL, the proliferation marker KI67, and the pluripotency marker OCT4 (Fig. 4B). Pre-induction iPSCs were positive for OCT4 and KI67, and did not show any expression of DDX4 or DAZL. Staining cells after DDX4+ cell induction and prior to sorting, we observed cells strongly expressing DAZL and DDX4, with the expected cytoplasmic localization. Most cells were still KI67+, and some DAZL- DDX4- cells retained OCT4 expression. The overall percentage of DAZL + DDX4+ cells by immunofluorescence was 20–25% (Fig. EV4A). Sorted DDX4+ cells expressed DDX4, DAZL, and KI67, but not OCT4. After 14 days of culture, DDX4+ cells retained expression of these markers (Figs. 4B and EV4A) and had increased in number (Fig. EV4B). We also stained for the PGC marker SOX17, and found that it was expressed only in PGCLCs, and not in iPSCs nor DDX4+ cells (Fig. EV4C). In this experiment, we also validated DDX4 expression using a separate antibody. Finally, we stained for the meiosis markers SYCP3 and RAD51, and found that they are not expressed in our DDX4+ cells (Fig. EV4D), thus distinguishing our DDX4+ cell induction protocol from a related protocol developed to generate cells that enter into meiosis and do not proliferate (Pierson Smela et al, 2024). Based on our live imaging (Fig. 4A) and immunofluorescence staining (Fig. 4B), we therefore conclude that our TF-derived DDX4+ cells are induced oogonia-like cells (iOLCs) based on the expression of mitotic oogonia markers (DDX4, DAZL, and KI67) and lack of earlier primordial germ cell and pluripotency markers (OCT4 and SOX17) and lack of later meiotic markers (SYCP3 and RAD51) and are capable of feeder-free expansion culture while retaining this oogonia-like identity post-sorting.

## Transcriptomic analysis of iOLCs shows similarities with human fetal germ cells and xrOvary oogonia

We next profiled the gene expression in iOLCs to provide a more robust phenotypic characterization and evaluate their similarity to

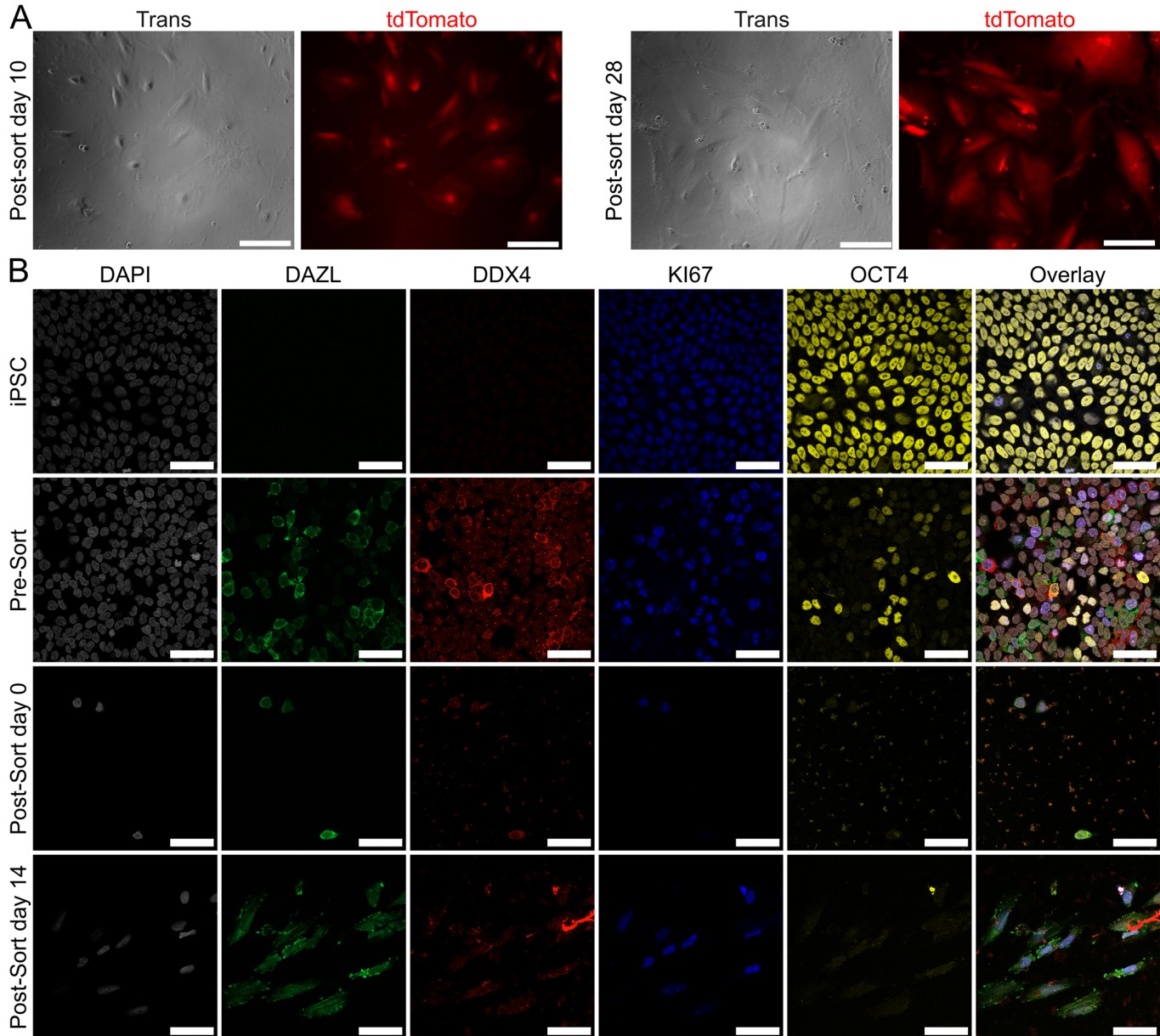

**Figure 4. iOLCs are capable of post-isolation feeder-free maintenance and expansion and display continued DDX4 protein expression.**

(A) Live cell fluorescent imaging is performed on FACS-isolated D3 combination-derived iOLCs grown feeder-free on matrigel-coated plates at day 10 and 28 post-FACS isolation. Transmission light microscopy is used to visualize cells, and tdTomato fluorescent protein derived from the cleaved DDX4-T2A-tdTomato reporter is visualized using the Texas Red channel of an epifluorescence microscope. Scale bars are 90 µm. (B) Staining of fixed cells for DAPI, DAZL, DDX4, KI67, and OCT4. Samples include pre-differentiation iPSCs, unsorted cells after 5 days of iOLC differentiation with the D5 combination, iOLCs fixed immediately after sorting for DDX4+ cells, and sorted iOLCs after 14 days of culture. Scale bars are 50 µm. Source data are available online for this figure.

other in vivo and in vitro analogous cell types. To investigate the effects of different iOLC induction methods, we induced iOLCs using our original protocol (D3 TFs without DNMT1 inhibition) and improved protocol (with DNMT1 inhibition, and either D3, D3 + ZGLP1, D3 + ANHX, or D5 TF combinations). Additional samples included iOLC TFs in combination with a constitutively active androgen receptor splice variant (AR-V7). We included AR-V7 because it had increased the yield of DDX4+ cells in our screening experiments (Fig. 3), and we were also interested to see if

this would upregulate prospermatogonia-related genes, since AR is expressed in a subset of prospermatogonia in vivo (Garcia-Alonso et al, 2022). We harvested iOLCs from each condition and performed bulk RNA-seq on the sorted DDX4+ cells. For the D5 TF combination, we also sequenced RNA from the unsorted cell population.

First, we examined the expression of known marker genes for different stages of germline development: pluripotent stem cells, primordial germ cells, oogonia, meiotic germ cells, and oocytes

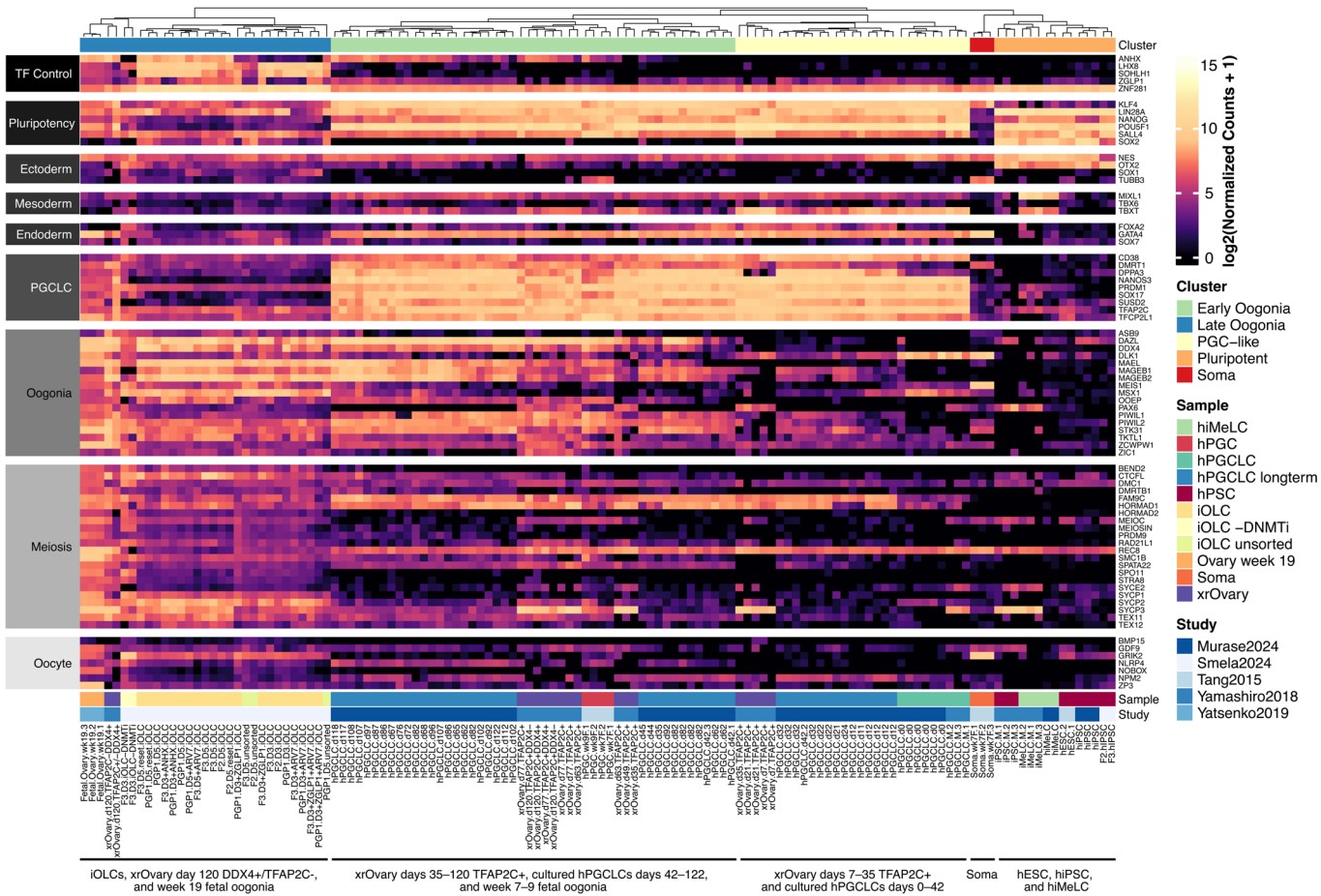

**Figure 5. Gene expression analysis of iOLCs and samples from other studies.**

Marker gene expression heatmap and unsupervised hierarchical clustering (UHC). From left to right, samples include: hiPSCs, iOLCs generated with D3 TFs and without DNMT1 inhibition, iOLCs generated with D3 and D5 TF combinations using the improved protocol; Tang et al, hESCs, fetal ovarian somatic cells, and female hPGCs; Yatsenko et al, week 19 fetal ovarian germ cells; and Yamashiro et al, hiPSCs, hiMeLCs, hPGCLCs, and xrOvary germ cells ordered by developmental stage. Samples are labeled at the bottom of the heatmap. Clusters found by UHC are labeled at the top.

(Fig. 5; Dataset EV3). We also included marker genes for off-target lineages including endoderm, mesoderm, and ectoderm. As a control, we used previously published data from human fetal germ cells and xrOvary oogonia (Tang et al, 2015; Yamashiro et al, 2018; Yatsenko et al, 2019). We found that iOLCs from all induction protocols expressed the oogonia marker genes *DDX4, DAZL, STK31,* and *MAEL*. However, iOLCs from the original protocol retained expression of the pluripotency/PGC marker *POU5F1,* and lacked expression of other oogonia markers such as *MAGEB1/2, MSX1, TKTL1,* and *OOEP*. By contrast, iOLCs from the improved protocol expressed these oogonia markers and had lower expression of *POU5F1,* similarly to xrOvary day 120 DDX4 + TFAP2C- oogonia. Additionally, iOLCs upregulated some meiotic markers, including *CTCFL, SYCP1/2/3, HORMAD1,* and *TEX11/12*. However, these cells were likely not fully committed to meiosis, as they had lower expression of other markers such as *REC8* and *SPO11,* and lacked protein expression of SYCP3 and RAD51 (Fig. EV4D). Similarly, iOLCs did not express genes related to oocyte growth, such as *NOBOX, GDF9, BMP15, ZP3,* and *NPM2*.

To compare our iOLCs with ovarian cells in vivo and with oogonia-like cells produced using other in vitro methods, we performed unsupervised hierarchical clustering (UHC) based on lineage-specific marker gene expression. Based on the UHC dendrogram, we identified five main clusters (Fig. 5). The first cluster, which we call late oogonia, has high expression of oogonia marker genes and low expression of pluripotency and PGC genes. This cluster contains our iOLCs, week 19 fetal ovarian cells (Yatsenko et al, 2019) and day 120 DDX4+/TFAP2C- xrOvary cells (Yamashiro et al, 2018). The second cluster, which we call early oogonia, has expression of both oogonia and pluripotency/PGC markers, and contains DDX4+/TFAP2C+ xrOvary cells as well as week 7 and 9 fetal germ cells (Tang et al, 2015) and day 42–120 cultured hPGCLCs (Murase et al, 2024). The third cluster contains hPGCLCs and early (day 0–42) cells from other in vitro oogonia induction methods. The fourth cluster contains ovarian somatic cells, and the fifth cluster contains pluripotent stem cells. Overall, UHC shows that our iOLCs correspond to late-stage oogonia, similar to DDX4+/TFAP2C- cells produced in xrOvary-based

methods (Yamashiro et al, 2018) and more mature than cultured hPGCLCs (Murase et al, 2024).

As an alternative approach to compare the whole transcriptome of our cells with in vivo data, we performed CellTypist annotation using a previously published reference atlas of the human fetal ovary for model training (Garcia-Alonso et al, 2022). This atlas contains multiple germ cell and somatic cell types from different developmental stages, including PGCs, gonadal germ cells, pre-meiotic and meiotic oogonia, oocytes, mesenchymal, pre-granulosa and granulosa cells, and several types of immune cells. Cell type annotation revealed that our iOLCs were more similar to pre-meiotic and meiotic oogonia in their overall gene expression, whereas oogonia-like cells generated by other methods (Murase et al, 2024; Yamashiro et al, 2018) were more similar to PGCs (Appendix Fig. S4A). We did not observe the similarity between iOLCs and off-target somatic lineages, except for macrophages, which were annotated as being similar to all samples including in vivo germ cells and hiPSCs. Furthermore, UMAP analysis demonstrated that D3 and D5-derived iOLCs from DNMTi treatment were readily clustered with Week 19 Fetal oogonia (Appendix Fig. S4B). Interestingly, the D3-derived iOLCs without DNMTi clustered more readily with recently described in vitro oogonia samples from Murase et al, 2024 (Appendix Fig. S4B). DGE and GO term enrichment demonstrated that DNMTi significantly enriched gene expression related to meiosis, cell cycle, recombination, and development (Appendix Fig. S5).

Next, we performed a differential gene expression analysis to identify differences among iOLCs induced using different TFs. In comparison to the D3 TF combinations, the other TF combinations showed significant changes in gene expression (Dataset EV4). We conducted a GO term enrichment analysis on differentially expressed genes. AR-V7 overexpression did not result in activation of spermatogenesis-related genes; the only enriched GO term among upregulated genes was "chromatin organization", and contrary to our expectations, "male sex differentiation" was enriched among downregulated genes. We found that overexpression of ANHX (relative to D3 control) upregulated genes related to reproductive development (including "spermatogenesis", "sexual reproduction", "gamete generation", "meiotic cell cycle", "developmental process involved in reproduction" and "retrotransposon silencing"), and downregulated genes related to mesenchymal and cardiac development. For overexpression of ZGLP1, no GO terms were significantly enriched in the differentially expressed genes. However, the D5 combination (overexpression of both ANHX and ZGLP1) showed more extensive changes. Upregulated genes were enriched for GO terms related to metabolism, meiosis, germ cell development, and oogenesis (Dataset EV5). GO terms enriched among downregulated genes were mainly related to cell adhesion, migration, blood vessel development, and angiogenesis. These results indicate that the two additional factors in the D5 combination act to reinforce germ cell development and down-regulate off-target differentiation.

We additionally compared gene expression in D5 DDX4+ iOLCs with unsorted cells (largely DDX4-) from the same induction. Relative to the iOLCs, the bulk population expressed less DDX4 (as expected), and also had lower expression of other oogonia markers DAZL, BOLL, TEX15, and STK31. Significantly downregulated GO terms were related to gametogenesis, meiosis, and metabolism (Dataset EV5). GO terms upregulated in the bulk

population relative to iOLCs were related to vascular, cardiac, and nervous development. We also compared gene expression in D5 DDX4+ iOLCs generated from primed hiPSCs vs. transient naïve reset hiPSCs. There were only a few significant differentially expressed genes (one downregulated and seven upregulated). However, we note that both SOX17 and GATA4 were significantly upregulated in iOLCs derived from reset hiPSCs, and these transcription factors are known to be expressed in fetal germ cells (Irie et al, 2015; Ketola et al, 2000).

For a closer look at similarities and differences between our D5 iOLCs and hPGCs and xrOvary day 120 oogonia, we performed a differential gene expression and GO term enrichment analysis. As a control, we also compared our iOLCs with fetal ovarian somatic cells from the Tang et al, dataset, and found that GO terms related to germ cells and meiosis (e.g., "meiotic cell cycle", "sexual reproduction", and "gamete generation") were significantly enriched among genes that were less expressed in fetal somatic cells relative to iOLCs. By contrast, we did not observe significant enrichment of GO terms related to germ cell development between iOLCs and hPGCs or xrOvary oogonia. Differentially expressed genes were mainly associated with metabolic processes such as "nitrogen compound metabolic process", or generic terms such as "multicellular organismal process" (Dataset EV5). However, terms related to meiosis, such as "negative regulation of mitotic cell cycle", "meiotic cell cycle", "DNA damage response", and "double-strand break repair via homologous recombination" were upregulated in xrOvary day 120 oogonia relative to our iOLCs. This may reflect that xrOvary day 120 oogonia are beginning meiosis, whereas our iOLCs are at an earlier developmental stage. Although these results should be incorporated with caution given the possibility of incomplete batch correction, it appears that our iOLCs express broadly similar levels of germ cell-related genes compared with hPGCs or xrOvary oogonia.

## iOLCs generated using the improved protocol show robust DNA methylation erasure similar to in vivo oogonia

During germline development, cells proliferate in the absence of maintenance DNA methyltransferase activity, leading to genome-wide erasure of DNA cytosine methylation (Tang et al, 2016). This is a crucial process for mammalian gametogenesis, as it erases parental imprints and allows the establishment of sex-specific imprinting. Furthermore, crucial genes for germ cell development, such as DAZL, are expressed from methylation-sensitive promoters (Linher et al, 2009; Zhang et al, 2016). Notably, previous TF-driven methods for generating oocyte-like cells from mouse pluripotent stem cells did not achieve DNA methylation erasure nor expression of Dazl (Hamazaki et al, 2021). However, previous xrOvary-based methods for generating human oogonia-like cells were able to erase DNA methylation over a culture period of 77–120 days (Yamashiro et al, 2018).

Given the shorter culture period (4–5 days) of our method, we investigated whether our iOLCs were also erasing their methylation. We performed whole genome enzymatic methyl sequencing on a total of 18 samples, including primed hiPSCs, transient naïve reset hiPSCs, hPGCLCs, and iOLCs generated using our original and improved protocols (Vaisvila et al, 2021). Notably, the improved protocol included a chemical inhibitor of maintenance DNA methyltransferase

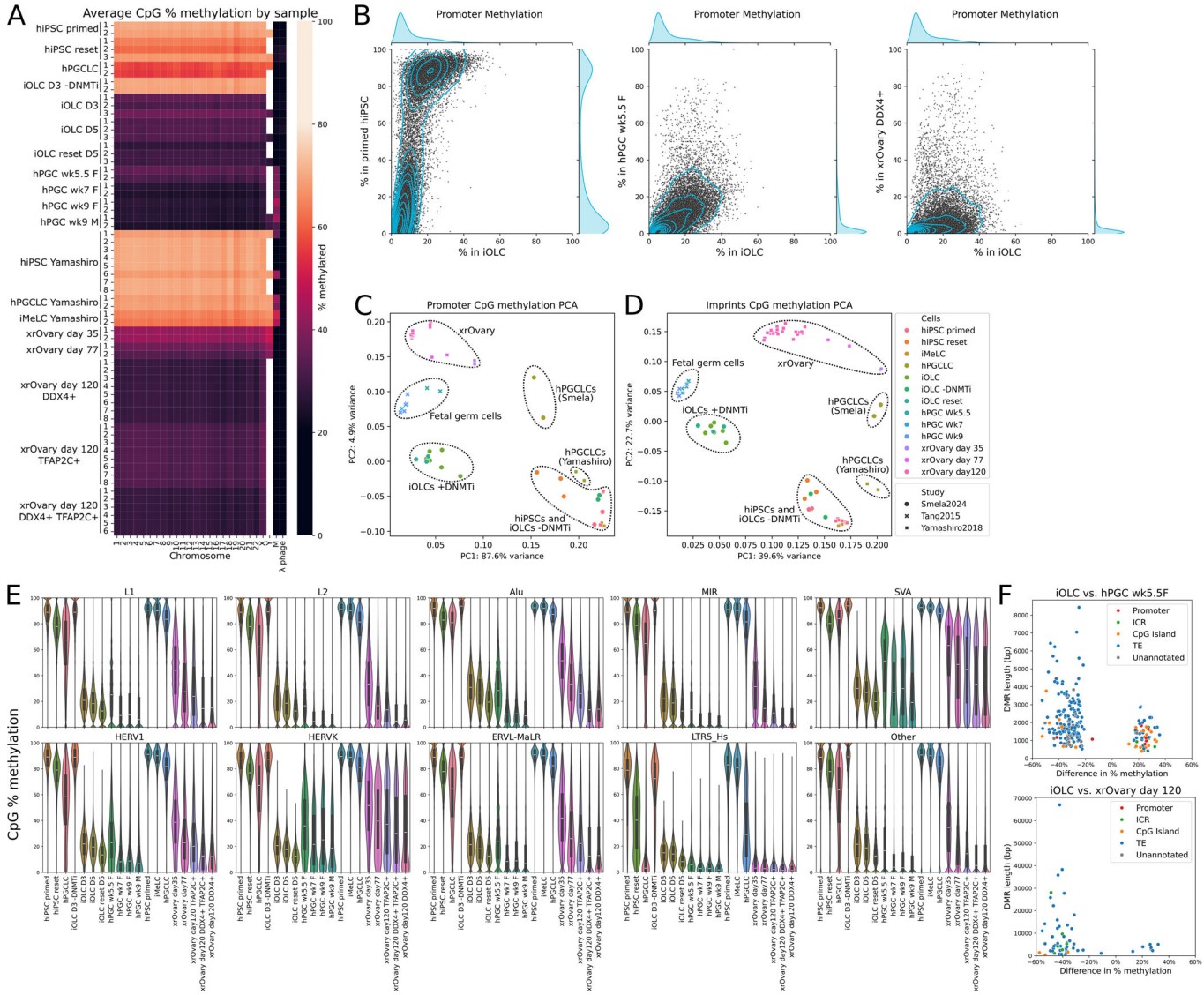

**Figure 6. Epigenetic analysis of iOLCs in comparison to hPGCs and xrOvary DDX4+ cells.**

For all panels, sample types and the number of biological replicates are as follows: primed hiPSC, $n = 2$; naïve reset hiPSC, $n = 3$; hPGCLC, $n = 2$; iOLC D3 without DNMTi, $n = 2$; iOLC D3, $n = 3$; iOLC D5, $n = 3$; iOLC reset D5, $n = 3$; hPGC female week 5.5, $n = 2$; hPGC female week 7, $n = 2$; hPGC female week 9, $n = 2$; hPGC male week 9, $n = 2$; Yamashiro et al, hiPSC, $n = 8$; Yamashiro et al, himeLC, $n = 2$; Yamashiro et al, hPGCLC, $n = 2$, Yamashiro et al, xrOvary day 35, $n = 2$, Yamashiro et al, xrOvary day 77, $n = 2$; Yamashiro et al, xrOvary day 120 DDX4+, $n = 8$; Yamashiro et al, xrOvary day 120 TFAP2C+, $n = 8$; Yamashiro et al, xrOvary day 120 DDX4 + TFAP2C+, $n = 6$. (A) Heatmap showing average % CpG methylation for each chromosome in each sample. (B) Comparing average methylation for each promoter in iOLCs vs. primed hiPSCs, hPGCs (female week 5.5), and xrOvary day 120 DDX4+ cells. Promoter methylation is efficiently erased in iOLCs. (C) Principal component analysis for promoter methylation among different sample types. Along PC1 (which captures 87.6% of variance), iOLCs are similar to hPGCs and xrOvary cells. (D) Principal component analysis for imprint methylation among different sample types. iOLCs are most similar to hPGCs. (E) Distribution of methylation for different classes of TEs among different sample types. In the violin plots, the center represents the median, bars represent quartiles, and whiskers represent minima and maxima. (F) Differentially methylated regions between iOLCs and hPGCs, and between iOLCs and xrOvary day 120 DDX4+ cells. Regions are colored based on annotated features. Most correspond to TEs.

DNMT1. We first analyzed overall methylation levels (Fig. 6A), comparing our samples to previously published in vivo hPGC data as well as data from the xrOvary culture system (Tang et al, 2015; Yamashiro et al, 2018). We observed that iOLCs from the original protocol, which did not include a DNMT1 inhibitor, did not have lower DNA methylation relative to hiPSCs. hPGCLCs showed a slight decrease in methylation. Encouragingly, iOLCs from the new protocol were strongly demethylated, reaching levels similar to xrOvary and hPGCs.

We next examined DNA methylation at functionally relevant loci including promoters (Fig. 6B,C), CpG islands (Fig. EV5A), imprinting control regions (ICRs) (Fig. 6D), and transposable elements (TEs) (Fig. 6E). In hiPSCs, promoters and CpG islands displayed bimodal methylation (Figs. 6B and EV5A). Most (56% of promoters and 61% of CpG islands) had a low (0–20%) level of methylation, and a smaller fraction (24% of promoters and 29% of CpG islands) had a high level (80–100%). By contrast, iOLCs had no promoters or CpG islands above 80% methylation, and the

proportion with lower than 20% methylation was greatly increased (73–92%). Notably, the methylation of the *DAZL* promoter decreased from 96% in hiPSCs to 26% in iOLCs. To evaluate the epigenetic similarity of our iOLCs with xrOvary cells and hPGCs, we performed a principal component analysis of promoter methylation. Along principal component 1, which captured most of the variance, our iOLCs were similar to xrOvary oogonia and hPGCs (Fig. 6C).

Erasure of parental methylation at ICRs is crucial for the generation of functional gametes (Tang et al, 2016). We examined DNA methylation at 332 candidate human ICRs found by a previous study (Jima et al, 2022) which identified regions with 50 ± 15% methylation in somatic tissues and 0 or 100% methylation in eggs and sperm. Interestingly, we found that these regions had an average of 76% methylation in our hiPSC samples. Previous studies in mouse and human iPSCs identified aberrant ICR hypermethylation as a consequence of iPSC reprogramming and this may have also occurred in our hiPSCs (Bar et al, 2017; Yagi et al, 2019). However, in iOLCs, only 12 out of 327 ICRs retained >40% methylation, with the majority having <15% methylation. A principal component analysis showed that our iOLCs were more similar than hiPSCs or xrOvary oogonia to in vivo hPGCs (Fig. 6D). Therefore, despite the relatively short duration of the iOLC induction protocol, iOLCs largely completed the process of imprint erasure.

Although genome-wide DNA demethylation is necessary for gametogenesis, demethylation of TEs can activate the expression of these potentially deleterious elements. In the human germline, certain TEs retain DNA methylation despite genome-wide demethylation (Tang et al, 2015). This "escape" from demethylation ensures that the TEs remain suppressed. Given that our protocol included a small-molecule inhibitor of DNMT1, we wondered if any TEs would escape demethylation. Examining methylation levels at 3,213,373 TEs, we found that most were demethylated in iOLCs (average methylation 22%) (Fig. 6E). However, we identified 280 "escapee" TEs that retained >80% methylation, and 8657 TEs which retained >60% average methylation. Thus, although 5 days of GSK-3484862 treatment was sufficient to erase DNA methylation at promoters and imprinting control regions, a subset of TEs remained methylated. Interestingly, this set of escapee TEs showed little overlap with escapee TEs observed in vivo or in xrOvary oogonia (Fig. EV5B).

Finally, we identified and annotated differentially methylated regions (DMRs) between our iOLCs and xrOvary and fetal oogonia. Among the in vivo germ cells (Tang et al, 2015), our iOLCs were most similar to week 5.5 female samples. There were 58 regions significantly more methylated than in vivo, and 173 loci significantly less methylated (Fig. 6F). Most DMRs were associated with TEs (154 regions), and only a small minority were associated with promoters (8 regions) or imprint control regions (eight regions). Among the xrOvary cells, our iOLCs were most similar to day 120 DDX4-positive cells. There were a total of seven regions where iOLCs were significantly more methylated than xrOvary cells, and 54 significantly less methylated. Again, most of these DMRs (54 regions) were associated with TEs, and in this case none were associated with promoters. By contrast, iOLCs generated using the protocol without DNMT1 inhibition had 208,715 DMRs, all of which were more methylated. Overall, our data indicate that five days of DNMT1 inhibition effectively reset DNA methylation

to an oogonia-like state except at a few regions, most of which are associated with TEs.

# Discussion

In this study, we establish TF-based differentiation methods to rapidly and efficiently produce human germ cell types. We computationally identify TFs with potential regulatory roles in human oogenesis, and screen these via combinatorial and individual overexpression in germline-reporter hiPSC lines. Based on the screening results, we find distinct sets of TFs that can produce human DDX4+ oogonia-like cells and hPGCLCs. We characterize our hPGCLCs and iOLCs and show that they are similar to the corresponding developmental stage of germ cells in vivo, as well as similar to previous xrOvary-based methods. These results will be particularly useful for producing oogonia-like cells, since existing methods require several months of cell culture (Sasaki et al, 2015; Yokobayashi et al, 2017; Yamashiro et al, 2018; Irie et al, 2015; Murase et al, 2024).

We demonstrate that overexpression of *DLX5*, *HHEX*, and *FIGLA* improves on-target canonical hPGCLC formation, shown in the expression of the core primordial germ cell genes *SOX17*, *TFAP2C*, and *PRDM1* and proteins SOX17, OCT4, ITGA6, and EpCAM in floating aggregate differentiation methods. While the overall hPGCLC yield, particularly in the no TF controls, was generally low with the D4TN3V reporter line utilized in this study, this yield was in line with the reported yields seen in other studies which have noted a wide variety in absolute hPGCLC yield between different hiPSC lines (Yokobayashi et al, 2017). Interestingly, overexpression of *DLX5*, *HHEX*, and *FIGLA* improved the hPGCLC yield to a level that is similar to high-performing hiPSC lines noted in other studies, and may represent a novel method for improving hPGCLC yield in difficult, low-yielding cell lines (Sasaki et al, 2015; Yokobayashi et al, 2017; Yamashiro et al, 2018; Irie et al, 2015). More studies will be needed to determine if these three TFs perform germ cell regulating roles in vivo in addition to serving as useful cell engineering tools in our described context. It is unclear from our system whether this is the canonical function of the TFs normally during hPGCLC formation or a result of forced overexpression. DLX5, a known amnion marker, for example, has been described in its expression in both in vivo hPGCs and in vitro-derived hPGCLCs (Chen et al, 2019; Sybirna et al, 2020; Fang et al, 2022). Differential expression of HHEX, a known endoderm marker, and DLX5 was noted in early day 1 and 2 doxycycline induction of SOX17, TFAP2C, and PRDM1 in a recent work describing the TF control of hPGCLC specification (Kojima et al, 2021). Likewise, our analysis of the fetal gonad atlas demonstrates the expression of both DLX5 and HHEX in various stages of germline development (Garcia-Alonso et al, 2022). For *FIGLA*, in particular, it is likely that our overexpression drives a non-canonical function that is not normally seen during hPGC formation, as *FIGLA* is not normally expressed at PGC developmental timepoints in vivo (Li et al, 2017). Nonetheless, this study motivates the utilization of our toolkit in a diverse range of hPGCLC screening modalities, which may uncover the role of new TFs in regulating gametogenesis and further elucidate the underlying gene regulatory network of primordial germ cell specification.

We show that overexpression in combination with *SOHLH1*, *LHX8*, and *ZNF281* (D3) drives the formation of DDX4+ cell

formation, termed iOLCs, in just four days. In this study, we demonstrate these TF-derived iOLCs express key mitotic oogonia markers (DDX4, DAZL, KI67), while lacking earlier primordial germ cell and pluripotency markers (SOX17, OCT4), and later meiotic markers (SYCP3, RAD51). Our DDX4+ cell formation is rapid compared to existing methods which generally require ~70 to 120 days in xrOvary-based methods compared to our 4-day protocol (Yamashiro et al, 2018). In addition, our DDX4+ cells are generated feeder-free using a 2D monolayer culture protocol, making scaling for production and screening simple and amenable to integration with current high-throughput screening designs. Additional studies will be needed to determine the in vivo role of these TFs in oogonia and oocyte formation and whether our overexpression-based method for generating these cell types recapitulates the endogenous genetic regulation of oogenesis. As ZNF281, LHX8, and SOHLH1 are highly expressed during human folliculogenesis, linked as causal genetic determinants of infertility when mutated, and in the case of, Lhx8 and Sohlh1, found in mouse models to induce oocyte-like formation from stem cells, it is highly likely they play critical roles during in vivo oogenesis (Pangas et al, 2006; Zhang et al, 2018b; Hamazaki et al, 2021). Expression of the additional TFs ANHX and ZGLP1 (in the D5 combination) increases the yield of iOLCs, upregulates genes related to oogenesis, and downregulates genes related to off-target cell types. ANHX is known to be expressed in oogonia, but its functional role has not yet been previously characterized (Garcia-Alonso et al, 2022). In mouse germ cells, Zglp1 overexpression is known to induce meiosis (Nagaoka et al, 2020), but in human germ cells, ZGLP1 is expressed in oogonia that are not yet activating meiosis (Garcia-Alonso et al, 2022). Via transcriptomic analysis, we find that our TFs drive the formation of DDX4+ cells that exhibit upregulation of key oogonia genes, which led us to designate these cells as induced oogonia-like cells. We find our iOLCs broadly share similarity with in vivo fetal oogonia (Zhang et al, 2018b; Li et al, 2017). Notably, our iOLCs have gene expression corresponding to a later developmental stage than other methods of generating oogonia-like cells (Fig. 5) (Murase et al, 2024), including lower expression levels of pluripotency and PGC marker genes. We also demonstrate continued culture feeder-free post-isolation of our iOLCs using media developed for hPGCLC maintenance, potentially expanding the utility of this cell type for long-term studies (Kobayashi et al, 2022).

In our optimized protocol for iOLC production, we found that inhibition of DNMT1 efficiently erases DNA methylation and leads to an epigenetic state closely resembling that of human fetal germ cells. Additionally, we find that DNMTi promotes the expression of meiosis and development-related gene pathways and results in iOLCs with more transcriptomic resemblance to in vivo oogonia (Appendix Figs. 4 and 5). We took advantage of a recently described noncovalent DNMT1 inhibitor, which lacks the toxicity of previous inhibitors such as 5-azacytidine (Haggerty et al, 2021; Azevedo Portilho et al, 2021). DNMT1 inhibition results in genome-wide demethylation through a lack of maintenance methylation during cell divisions, similar to the process that takes place in the human germline (Tang et al, 2016). In mouse germ cells, Dnmt1 is largely blocked from maintaining genome-wide methylation, but retains activity at specific loci related to transposable elements, imprinting control regions, and promoters of meiotic genes (Hargan-Calvopina et al, 2016). These loci are actively demethylated by Tet enzymes prior to meiotic initiation. Notably, conditional knockout of Dnmt1 in mouse PGCs results in precocious expression of Dazl and differentiation to oocytes and prospermatogonia (Hargan-Calvopina et al, 2016). Chemical inhibition of DNMT1 in our protocol may have similar effects. Although we did observe some TEs escaping DNA demethylation in our iOLCs, it is possible that this reflects de novo methylation by DNMT3. If DNMT1 retains activity in vivo at TE-associated loci, this could explain the TE-associated differential methylation we observed between in vivo hPGCs and our iOLCs.

More research is needed to determine the functionality of these iOLCs and whether they can contribute to further gametogenesis and oocyte formation, such as by entering and performing meiosis and oocyte maturation. Furthermore, additional studies of dosage control and the relative contribution of each TF for driving and modulating differentiation in order to provide a more granular dissection of the underlying TF network are needed, as this study utilized polyclonal cell lines harboring multiple, transposon based insertions of each TF making this analysis infeasible. Further research combining our iOLC methods with co-culture methods such as xrOvaries (Yamashiro et al, 2020) or newer all-human methods such as those described recently (Pierson Smela et al, 2023a) are needed to determine if these DDX4+ cells are capable of meiotic entry or oocyte maturation. Nonetheless, our rapid and direct generation of human iOLCs provides a novel, reproducible, and simple-to-obtain source of important germline models that can be used in reproductive development and genetic screening studies as models of inaccessible in vivo counterparts. In conclusion, we identify novel TFs in driving germ cell formation, establishing a rapid, high-throughput platform for in vitro human gametogenesis and reproductive modeling in a simple and efficient manner.

## Methods

**Reagents and tools table**

| Reagent/Resource | Reference or source | Identifier or catalog number |
| --- | --- | --- |
| **Experimental models** | | |
| F3 hiPSCs | ATCC | BXS0116 |
| F2 hiPSCs | ATCC | BXS0115 |
| PGP1 hiPSCs | Personal genome project | PGP1 |
| Rhesus iPSC Line 80 | Amander Clark | Line 89 |
| Rhesus iPSC Line 90 | Amander Clark | Line 90 |
| **Recombinant DNA** | | |
| PB-cT2G-cERP2 | Addgene | 175503 |
| dCas9-KRAB-meCP2 | Addgene | 110821 |
| **Antibodies** | | |
| *Flow Cytometry* | | |
| CD38 PerCP-Cy5.5 Mouse IgG | Biolegend | 303522 |
| EpCAM-APC-Cy7 Mouse IgG | Biolegend | 324245 |
| Integrin-alpha-PE Rat IgG | Biolegend | 313607 |

| Reagent/Resource | Reference or source | Identifier or catalog number |
|---|---|---|
| *Immunofluorescence Primary* | | |
| OCT4- Mouse IgG | Abcam | AB398736 |
| DDX4- Mouse IgG | Abcam | AB27591 |
| DAZL- Rabbit IgG | Abcam | AB215718 |
| SOX17- Goat IgG | Abcam | AB355060 |
| ITGA6- Rat IgG | Abcam | AB493634 |
| *Immunofluorescence Secondary* | | |
| Mouse IgG-AF647 Donkey | Abcam | AB162542 |
| Rabbit IgG-AF488 Donkey | Abcam | AB2534104 |
| Goat IgG-AF568 Donkey | Abcam | AB2534104 |
| **Oligonucleotides and other sequence-based reagents** | | |
| DNMTi sgRNA 1: GGCTTCAGCAGACGCGGCGG | DNMT1 promoter | NA |
| DNMTi sgRNA 2: GGTACGCGCCGGCATCTCGG | DNMT1 promoter | NA |
| DNMTi sgRNA 3: GGAGGCTTCAGCAGACGCGG | DNMT1 promoter | NA |

## Ethics statement

All experiments on the use of hiPSCs for the generation of hPGCLCs and oogonia-like cells were approved by the Embryonic Stem Cell Research Oversight (ESCRO) Committee at Harvard University.

## Gene regulatory network inference and TF prioritization

To perform the PageRank TF prioritization method the following steps were followed:

- RNA-seq datasets were obtained from the Gene Expression Omnibus (GEO) database, and log2fc values for each aligned gene for each sample were calculated using the DESeq2 package (Love et al, 2014)
- Gene regulatory networks were inferred utilizing the GRNBoost2 algorithm in the Arboreto computational framework (Love et al, 2014; Moerman et al, 2019)
- PageRank was calculated for each transcription factor in the resulting network via the NetworkX package, and ranked factors were visualized using Seaborn. The full code and corresponding Jupyter notebooks for the standard STAMPScreen pipeline can be found at: https://github.com/programmablebio/stampscreen (Kramme et al, 2021a)

  Gene Ontology Analysis was performed by:
- Input of the prioritized TF screening list to the g:profiler tool and selecting the significantly enriched BP terms. g:GOSt performs functional enrichment analysis, also known as over-representation analysis (ORA) or gene set enrichment analysis, on the input gene list. It maps genes to known functional information sources and detects statistically significantly enriched terms.
- These significant GO terms, selected by the adjusted *p* value, were then input to the visualizer tool Revigo and plotted as a scatterplot. The axes in the plot have no intrinsic meaning. Revigo uses multidimensional

scaling (MDS) to reduce the dimensionality of a matrix of the GO terms pairwise semantic similarities. The resulting projection may be highly non linear. The guiding principle is that semantically similar GO terms should remain close together in the plot.

## Cell lines used

For these studies, we primarily used the ATCC-BXS0116 female hiPSC line, which we term F3. This cell line was determined to be karyotypically normal via the Karyostat assay and scored as normal in the Pluritest compared to the assays' control dataset (Thermo Fisher). For validation of iOLC production, we also used the ATCC-BXS0115 female hiPSC line (which we term F2), the PGP1 male hiPSC line, and male rhesus iPSC lines 89 and 90 (Sosa et al, 2016). The rhesus iPSC lines were a gift from Prof. Amander Clark.

## iPSC culturing

All hiPSCs were maintained feeder-free on hESC-qualified matrigel-coated plates (Corning), as the manufacturer suggested dilution. hiPSCs were maintained in mTeSR media, with mTeSR Plus utilized for standard expansion, passaging and cell line creation and mTeSR1 utilized prior to induction where specified. hiPSCs were passaged in mTeSR1 for at least one passage prior to differentiation in order to remove the stabilized FGF present in mTeSR Plus. Cells were passaged 1:10 to 1:20 every 3–4 days using Accutase and seeded in the presence of 10 µM Y-27632, with media being changed every day for mTeSR1 or every other day when mTeSR plus was utilized. Rhesus iPSCs were cultured on mitomycin-inactivated DR4 MEFs (ATCC #SCRC-1045) in PluriStem medium supplemented with 1 µM XAV939, and passaged using Accutase. Cells were regularly tested for mycoplasma contamination using the ATCC Universal Mycoplasma Detection PCR kit.

## Generation of gametogenesis cell-reporter hiPSC lines

To generate the reporter cell lines, the following process was followed:

- Homology arms for target genes (*DDX4, NPM2,* and *NANOS3*) were amplified by PCR from genomic DNA.
- For each gene, a targeting plasmid, containing an in-frame C-terminal T2A-fluorescent reporter of either *tdTomato* (for *DDX4*), mGreenLantern (for *NPM2*), or *mVenus* (for *NANOS3*), as well as a *Rox-PGK-PuroTK-Rox* selection cassette, was constructed by Gibson assembly. The plasmid backbone additionally had an MC1-DTA marker to select against random integration.
- sgRNA oligos targeting the C-terminal region of target genes were cloned into the pX330 Cas9/sgRNA expression plasmid (Addgene 42230).
- For generation of the reporter line, 2 µg donor plasmid and 1 µg Cas9/sgRNA plasmid were co-electroporated into F3 hiPSCs, which were subsequently plated in one well of a six-well plate.
- Electroporations were performed using a Lonza Nucleofector with 96-well shuttle, with 200,000 hiPSCs in 20 µL of P3 buffer.
- Pulse setting CA-137 was used for all electroporations.
- Selection with the appropriate agent was begun 48 h after electroporation and continued for 5 days.

- After selection with puromycin (400 ng/mL), colonies were picked manually with a pipette.
- The hiPSC lines generated were genotyped by PCR for the presence of wild-type and reporter alleles.
- Homozygous clones were further verified by PCR amplification of the entire locus and Sanger sequencing.
- To excise the selection cassette, hiPSCs were electroporated with a plasmid expressing Dre recombinase.
- Selection was performed with ganciclovir (4 μM) and colonies were picked as described above. The excision of the selection cassette was verified by genotyping.
- Reporter lines were screened for common karyotypic abnormalities using a qPCR kit (Stemcell Technologies) followed by verification via Thermo Fisher Cell ID + Karyostat and Pluritest services.
- Additionally, whole genome sequencing was performed (Novogene, 10X coverage, 150 bp paired-end reads) to further verify the reporter alleles. Reads were aligned, analyzed, and visualized using the SeqVerify pipeline (version 1.1.0) (Pierson Smela et al, 2023b).

## cDNA vector creation

Vectors for cDNA overexpression were generated via MegaGate cloning, which was performed as follows:

- Full-length cDNAs for each TF of interest were either derived from the human ORFeome or synthesized as full-length constructs.
- All ORFs were cloned into pENTR221 with stop codons and minimal Kozak sequences.
- MegaGate was utilized to insert ORFs into the final PB-cT2G-cERP2 3' UTR barcode-modified expression vectors (Addgene 175503).
- Three unique barcodes were selected for each ORF with an average hamming distance of six.
- The three barcoded vectors for each ORF were then pooled, such that for each individual TF there was a mixture of three barcoded vectors.
- Sanger sequencing was performed across the entire ORF length to confirm canonical sequence with no amino acid changes.
- For *DNMT1* CRISPRi, the dCas9-KRAB-meCP2 repressor (Addgene 110821) (Yeo et al, 2018) was used. Three plasmids were constructed with different sgRNAs targeting the promoter of *DNMT1* (GGCTTCAGCAGACGCGGCGG, GGTACGCGCCGG CATCTCGG, GGAGGCTTCAGCAGACGCGG), and these were used as an equal-ratio mixture.

## Generation of inducible TF hiPSC cell lines

Expression plasmids containing TF cDNAs under the control of a doxycycline-inducible promoter were integrated into hiPSCs using piggyBac transposase, using the following process:

- To perform the integration, 100 fmol of TF cDNA plasmid, 200 ng piggyBac transposase expression plasmid, and 200,000 hiPSCs were combined in 20 μL of Lonza P3 buffer and electroporated using a Lonza Nucleofector 4D.
- Pulse setting CM-113 was used for all electroporations.
- After electroporation, cells were seeded in 24-well plates in mTeSR Plus + 10 μM Y-27632.

- Selection with 400 ng/ml puromycin began 48 h after electroporation and continued for 3–5 days.
- Cells were then passaged without drug selection for 3 days to allow for non-integrated plasmid loss.
- Finally, cells were again passaged under drug selection to generate a pure, polyclonal integrant pool.
- Presence and approximate copy number of integrated TF plasmids was confirmed by qPCR on genomic DNA.
- For hPGCLC and oogonia generation, polyclonal pools of hiPSCs were utilized.
- Average copy number was 8–10.
- The same procedure was performed for generating combinatorial cell lines, in which the 100 fmol of cDNA vector was divided equally between each TF for a pooled nucleofection.
- For copy number assessment the following was performed:
  - 1) RT-qPCR using SYBR Green master mix was performed after gDNA extraction using the DNAeasy kit.
  - 2) 10 ng of input gDNA was used per reaction based on the standard curve, with an anneal temperature of 60 degrees.
  - 3) To calculate copy number, the $2^{\Delta Cq+1}$ method was used, with RNAseP as a reference.
  - 4) The resultant value was multiplied by two to account for the two autosomal copies of *RPP30*.

## Generation of NANOS3+ and DDX4+ cells in 2D monolayer

For the generation of NANOS3+ and DDX4+ cells in monolayer, an identical induction format, and media composition was utilized:

- hiPSCs containing integrated TF expression cassettes were cultured in mTeSR1 medium on Matrigel-coated plates.
- For induction in monolayer, hiPSCs were dissociated to single cells using Accutase and seeded onto Matrigel-coated plates at a density of 2500–3000 cells per cm$^2$ in mTeSR1 + 10 μM Y-27632 and 1 μg/ml doxycycline for 6 h.
- Media was then removed and washed with DMEM/F12 and replaced with Media 1 (see components list below).
- After 12–18 h of induction, Media 1 was removed and washed with DMEM/F12 and replaced by Media 2.
- After 24 h, Media 2 was removed and replaced by Media 3.
- After 24 h, Media 3 was replaced with Media 4.
- Reporter-positive cells could then be harvested for use after 24 h in Media 4.
- Putative hPGCLCs could be isolated via the NANOS3 reporter expression, CD38 cell surface expression, combinations of both or EpCAM/ITGA6 dual positive cell surface markers.
- Induced oogonia-like cells (iOLCs) could be isolated via a DDX4 reporter.
- hPGCLCs and iOLCs could additionally be generated via embryoid body formation through methods established by Irie et al, 2015, Sasaki et al, 2015, Yamashiro et al, 2018, Kobayashi et al, 2022, and Murase et al, 2020.

Media formulations were as follows:
**Basal media (aRB27):**

- Advanced RPMI
- 1X B27 minus Vitamin A

- 1X Glutamax,
- 1X Non-Essential Amino Acids
- 10 µM Y-27632
- 1X Primocin or Pen-Strep
- 1 µg/ml Doxycycline.

**Media #1 (Epiblast-induction media):**

- aRB27 Basal Media (see above)
- 3 µM CHIR99021
- 100 ng/ml Activin A
- 0.1 µM PD173074

**Media #2:**

- aRB27 Basal Media (see above)
- 1 µM XAV939
- 40 ng/ml hBMP4

**Media #3:**

- aRB27 Basal Media (see above)
- 1 µM XAV939
- 100 ng/ml SCF
- 50 ng/ml EGF

**Media #4:**

- aRB27 Basal Media (see above)
- 1 µM XAV939
- 40 ng/ml hBMP4
- 100 ng/ml SCF
- 50 ng/ml EGF

## Fluorescent activated cell sorting of hPGCLCs and iOLCs

hPGCLCs and iOLCs were analyzed using flow cytometry on the BD LSRFortessa or Cytoflex LX machine, using the following process:

- Cell events were selected on FSC-A versus SSC-A, singlets are chosen via FSC-W versus FSC-H dispersion
- Live cells were selected based on DAPI negative gating on the Indo-Violet-A channel (Appendix Fig. S6).
- Negative gates were set using hiPSC controls or unstained cell controls.
- For any multi-staining flow cytometry, single stain controls and beads were utilized for compensation.
- For cell sorting for post-isolation growth, cells were captured using a Sony SH800S cell sorter.
- All flow cytometry analysis was conducted using the FlowJo software system (v10.8.1).

NANOS3 and DDX4 were captured via fluorescent reporters and constructed using the process described above. For cell surface markers, the following antibodies were utilized:

- CD38 PerCP-Cy5.5 Mouse IgG (Biolegend 303522)
- EpCAM-APC-Cy7 Mouse IgG (Biolegend 324245)
- Integrin-alpha-PE Rat IgG (Biolegend 313607).

For straining the following was performed:

- A 1:60 dilution of each antibody in a dPBS + 3% FBS FACS buffer was utilized.
- Staining was performed for one hour at 4 °C after harvesting with Accutase, and cells were sorted without fixation with additional DAPI staining.

For analysis of geometric mean fluorescent intensity in Fig. 2, the analysis was performed using Flowjo:

- The positive population was gated
- Using the reporter-positive population only, the geometric mean of the fluorescent intensity of the *tdTomato* reporter was plotted.

## Barcode enrichment screening for additional factors

Barcode enrichment screening was performed via the following process:

- Pooled PiggyBac plasmid libraries (Fig. 3A,B) were integrated into 1 million hiPSCs per replicate, using a Lonza nucleofector with 100 µL of P3 buffer.
- 250 fmol transposon pool and 2.5 µg transposase plasmid were used for each nucleofection.
- Puromycin selection was performed between days 2–7 post-nucleofection.
- Subsequently, DDX4+ iOLCs were induced using the protocol described above.
- In follow-up rounds of screening, cells were treated with 5 µM GSK-3484862, and some replicates were additionally treated with the retinoic acid receptor agonist AM580 (1 µM).
- DNA was extracted from sorted DDX4+ cells using the Qiagen DNEasy Micro kit, and from unsorted and pre-induction cells using the Qiagen DNEasy kit.
- Barcode amplification and sequencing was performed as previously described (Pierson Smela et al, 2023a).
- For fold-change calculations, barcode abundances in the DDX4+ cells were separately compared to unsorted cells and to pre-induction cells.

## Improved protocol for iOLC induction

For the improved protocol for iOLC induction, which yields a higher percentage of DDX4+ cells, the following was performed:

- Human or rhesus iPSCs were dissociated using Accutase and seeded onto Matrigel-coated plates at a density of 50,000/cm² in cold mTESR1 supplemented with:
  - 2% Cultrex (R&D Systems)
  - 10 µM Y-27632
  - 5 µM GSK-3484862
  - 1 µg/mL doxycycline.
- After 24 h, cells were washed with RPMI, and the medium was changed to aRB27 (described above) with:
  - 25 ng/mL hBMP4
  - 2% Cultrex
  - 5 µM GSK-3484862
  - 1 µg/mL doxycycline.

- After 24 h, the cells were fed with fresh medium.
- After 24 h, the medium was changed to aRB27 with:
  - 50 ng/mL SCF
  - 50 ng/ml EGF
  - 10 ng/mL hLIF
  - 5 µM GSK-3484862
  - 1 µg/mL doxycycline.

- After 24 h, the cells were fed with fresh medium.
- After 24 h, cells were harvested with Accutase for downstream analysis.

## Transient naïve resetting of hiPSCs

Naïve resetting was performed as previously described (MacCarthy et al, 2024). Briefly:

- 3 µg of episomal pCXLE-Sox2$^{A61V}$-2A-Klf4-2A-Myc (Addgene #210017) and 3 µg of pCXWB-EBNA1 (Addgene #37624) were co-nucleofected into $1 \times 10^6$ primed hiPSCs using the Lonza 4D-nucleofector ("Primary Cell P3" solution, "CM-113" program)
- Cells were plated on a feeder layer of irradiated CF1 MEFs (Thermo Fisher, $4 \times 10^6$ MEFs per six- well plate) at $8 \times 10^4$ nucleofected cells per well, in StemFlex media supplemented with 10 µM Y-27632.
- The next day, the medium was changed to PXL medium (described below).
- The cells were fed daily.
- At day 6, the cells were harvested using Accutase, and used for iOLC induction (without sorting).

Composition of PXL medium:

- 1:1 mix of Neurobasal medium (Gibco, 21103049) and Advanced DMEM/F12 (Gibco, 11320082) supplemented with:
  - 1X N2 (Gibco, 17502048)
  - 1X B27 minus vitamin A (Gibco, 12587010)
  - 1X sodium pyruvate (Gibco, 11360070)
  - 1X non-essential amino acids (Gibco, 11140050)
  - 1X GlutaMAX (Gibco, 35050061)
  - 1X Penicillin-Streptomycin (Gibco, 15070063)
  - 0.1 mM b-mercaptoethanol (Gibco, 31350010, 1 ml in 500)
  - 50 µg/ml L-ascorbic acid (Sigma, A8960)
  - 0.2% Geltrex (Gibco, A1413301)
  - 1 µM PD0325901
  - 2 µM XAV939 (Sigma, X3004)
  - 20 ng/ml hLIF (Peprotech, 300-05).

## Enzymatic methylation sequencing and analysis

For methylation sequencing preparation the following was performed:

- Genomic DNA was extracted from hiPSCs, hPGCLCs, and sorted DDX4+ iOLCs using the Qiagen DNeasy Micro kit.
- Libraries were prepared from 200 ng DNA using the NEB Enzymatic Methyl-Seq kit (NEB #E7120) following the manufacturer's instructions.

- Briefly, control DNA (methylated pUC19 and unmethylated λ phage) was spiked into each sample.
- Next, samples were sheared to ~300 bp using a Covaris S2 sonicator.
- Enzymatic methyl conversion was performed, and libraries were amplified using unique dual-index primers.
- 2 ×150 bp paired-end sequencing was performed on an Illumina NovaSeq.

For methylation analysis, the following was performed:

- Reads were aligned using bismark (v 0.24.0) (Krueger and Andrews, 2011) to the hg38 reference genome, with additional pUC19 and λ phage sequences for analysis of the control DNA.
- For analysis of methylation at functional loci, CpGtools CpG_aggregation.py (Wei et al, 2021) was used to aggregate methylation for features of interest, including promoters (defined as 900 bp upstream and 400 bp downstream of all transcriptional start sites), imprint control regions, CpG islands, and transposable elements.
- With the exception of imprint control regions (obtained from Jima et al, 2022) (Jima et al, 2022) genomic coordinates of all features were obtained from the UCSC hg38 TableBrowser database (https://genome.ucsc.edu/cgi-bin/hgTables) (Karolchik et al, 2004)
- Identification of differentially methylated regions was performed using the R package DSS (version 2.50.1) with default settings (Park and Wu, 2016)
- As DSS only calculates a false discovery threshold for individual loci, not regions, we performed a permutation analysis to identify a suitable areaStat threshold, which we set at 600.
- Annotation of differentially methylated regions was performed using `bedtools intersect` (version 2.31.0) (Quinlan and Hall, 2010) with the feature lists described above.
- For making plots, features were colored in reverse order of abundance (promoters > ICRs > CpG islands > TEs).

## Transcriptomic characterization of TF-induced cell lines and analysis

Cells were induced according to the above protocols and isolated for bulk RNA sequencing, which was performed as follows:

- Library preparation was performed using the NEB Next ultra-low input RNA-sequencing library preparation kit with a PolyA capture module for samples containing less than 10,000 cells.
- For samples with greater than 10,000 cells, the NEB Next Ultra II RNA-sequencing library preparation kit with PolyA mRNA capture module was utilized.
- Sequencing was performed on Illumina Next-Seq 500 and NovaSeq platforms with $2 \times 100$ bp or $2 \times 150$ bp paired-end reads, respectively.
- Only samples with RNA quality RIN scores of greater than 8 were utilized for analysis.

Sample derivation for analysis was performed as follows:

- For DLX5, HHEX, and FIGLA hPGCLC bulk RNA-sequencing, TFs were induced for 4 days and cells were sorted for NANOS3+ expression and utilized for bulk RNA-Seq.

- No-TF control cells were also utilized and sorted via NANOS3.
- For long-term culture (LTC) hPGCLCs, data was downloaded from GEO and realigned using our pipeline.
- For D3 DDX4+ bulk RNA-Seq, TFs were induced for 4 days and cells were sorted for DDX4+ expression and utilized for bulk RNA-Seq.
- For reference samples of the ovarian atlas, data was downloaded from GEO and realigned using our pipeline.
- The reference samples used are indicated in Dataset EV3.

RNA-seq analysis was performed as follows:

- Reads from our samples as well as previously published datasets were aligned to the human reference genome (GRCh38) using STAR (version 2.7.6) (Dobin et al, 2013) to construct count matrices aligning sequencing reads to the known set of human genes.
- For samples with $2 \times 150$ bp reads, the STAR options `--out-FilterScoreMinOverLread 0.33 --outFilter-MatchNminOverLread 0.33` were used in order to keep the minimum match length equivalent between the different samples.
- After alignment, reads were normalized using DESeq2's median of ratios (Love et al, 2014)
- A heatmap was generated in R (version 4.3.2) using Complex-Heatmap.
- Differential expression analysis was performed using DESeq2 (version 1.42.0).
- For comparisons between different studies, pluripotent stem cell samples (our hiPSCs, Yamashiro et al hiPSCs, and Tang et al hESCs) were labeled as "hPSCs", and each study was added as a covariate to the DESeq2 model.
- For each comparison, lists of significantly ($p_{adj} < 0.05$) upregulated ($\log_2 fc > 1$) and downregulated ($\log_2 fc < -1$) genes were used as input for GO term enrichment analysis with the PantherDB API (version 18.0) (Mi et al, 2021)

## Cell type annotation

Cell type annotation and classification was performed as follows:

- A reference single-cell RNA-seq (scRNA-seq) atlas was curated by merging 10,000 cells from the human germ cell lineages, 10,000 cells from the human somatic cell lineages, and 10,000 cells from the human immune cell lineages, all from the Reproductive Cell Atlas (https://www.reproductivecellatlas.org/gonads.html) into a combined `h5ad` file.
- Cells were filtered, normalized, and preprocessed in `scanpy` for input into CellTypist for fine-tuning (https://github.com/Teichlab/celltypist) via the `celltypist.train` function.
- Model training was conducted on 1xNVIDIA H100 GPU with 80 GB of VRAM for 6 GPU h, and the trained model was applied to the gene expression vector of each bulk RNA-seq sample, including both reference samples and study samples.
- Cell annotations are visualized on the heatmap generated as previously mentioned.

## Floating aggregate hPGCLC induction and harvesting

hiPSCs were differentiated from hPGCLCs according to the method of Sasaki et al, 2015 (Sasaki et al, 2015) Briefly:

- iMeLCs were induced by plating hiPSCs on matrigel-coated plates in GK15 medium, which is GMEM with:
  - 15% KSR
  - 0.1 mM NEAA
  - 2 mM L-glutamine
  - 1 mM sodium pyruvate
  - 0.1 mM 2-mercaptoethanol
- Further supplemented with:
  - Activin A
  - CHIR
  - ROCK inhibitor (Y-27632).
- The hPGCLCs were induced for four days by plating iMeLCs into a well of a low-cell-binding V-bottom 96-well plate in GK15 supplemented with:
  - LIF
  - BMP4
  - SCF
  - EGF
  - ROCK inhibitor

For PGCLC harvesting and staining for FACS, for each sample, the following was performed:

- Nine aggregates were combined in a 1.5 mL tube.
- The aggregates were washed with PBS, resuspended in trypsin solution (45 µL), and incubated in a shaking heat block (37 °C, 800 rpm).
- After 8 min, 940 µL FACS buffer (3% FBS in PBS) was added, and the aggregates were dissociated by vigorous pipetting.
- The suspension was passed through a 70 µm strainer and spun down ($300 \times g$, 3 min).
- The supernatant was removed, and the pellet was resuspended in antibody solution (40 µL).
- The suspension was kept on ice in the dark for 30 min.
- Then, 940 µL FACS buffer was added.
- The cells were spun down again and resuspended in 200 µL FACS buffer with 100 ng/mL DAPI, and analyzed on a BD LSR Fortessa flow cytometer as described above.

## iOLC maintenance culture

Feeder-free maintenance of iOLCs post-isolation was accomplished on Matrigel-coated plates with growth in S-CM media, established by Kobayashi et al, 2022, which is an STO-feeder cell conditioned media supplemented with SCF. The TLC expansion protocol is identical to the expansion protocol for the long-term culture of hPGCLCs and shows maintenance of DDX4+ expression over 28 days. The hPGCLC basal medium contained:

- Glasgow's MEM with:
  - 2 mM glutamine (Thermo Fisher, 11710035)
  - 13% (v/v) KSR
  - 1x NEAA
  - 1 mM sodium pyruvate
  - 1x penicillin-streptomycin

STO-CM was prepared by:

- Maintaining 5.0e6 mitomycin C treated STO cells in 12 mL of hPGCLC basal medium for 24 h,

- Removing cells by centrifugation
- Storing frozen at −20 °C until use.

The complete hPGCLC maintenance medium (S-CM for SCF-supplemented CM) was prepared by adding:

- 0.1 mM b-mercaptoethanol
- 50 mg/mL L-ascorbic acid
- 100 ng/mL recombinant human SCF to the CM.

FACS-enriched DDX4+ iOLCs (100–5000 cells) were inoculated onto Matrigel-coated plates in a well of six-well plates (Corning, 3506) with SCF-supplemented hPGCLC basal medium containing 10 μM Y-27632 + 1 μg/ml doxycycline. The medium was changed every other day without Y-27632. During the iOLC expansion, cells were dissociated using Accutase and passaged onto fresh matrigel every 5–14 days, when cells reached 50% confluency.

## Immunofluorescence imaging

For all 2D immunofluorescent imaging, the following was performed:

- Cells cultured on Matrigel-coated ibidi eight-well plates (iOLCs) (ibidi, cat 80806) were washed once with 200 μL dPBS and fixed by treatment with 200 μL of 4% paraformaldehyde in dPBS for 10 min at room temperature.
- The dPBS wash was repeated twice, and the cells were permeabilized by treatment with 200 μL 0.25% Triton X-100 in dPBS for 10 min at room temperature.
- The cells were washed with 200 μL PBST (0.1% Triton X-100 in dPBS), and blocked with 100 μL blocking buffer (1% bovine serum albumin and 5% normal donkey serum [Jackson ImmunoResearch, cat 017-000-121, lot 152961] in PBST) for 30 min at room temperature.
- The blocking buffer was removed and replaced with a solution of primary antibodies in the blocking buffer, and the cells were incubated overnight at 4 °C.
- The antibody solution was removed, and the cells were washed three times with 200 μL PBST.
- The cells were incubated with 100 μL secondary antibody solution in a blocking buffer for 1 h at room temperature in the dark.
- The secondary antibody was removed and replaced with 200 μL of DAPI solution (1 ng/mL in dPBS).
- After 10 min, the DAPI solution was removed, and the cells were washed twice with 200 μL dPBS, and stored in dPBS at 4 °C in the dark until imaging (typically a few hours).

For immunofluorescent imaging of floating aggregates, the following was performed:

- hPGCLC floating aggregates were washed with PBS and fixed with 1% PFA overnight at 4 °C.
- After another PBS wash, aggregates were detached from the Transwell.
- In preparation for cryosectioning, and transferred to 10% sucrose in PBS.
- After 24 h at 4 °C, the 10% sucrose solution was removed and replaced with 20% sucrose in PBS.

- After an additional 24 h at 4 °C, the ovaroids were embedded in the OCT compound and stored at −80 °C until sectioning.

For sectioning of the aggregates, the following was performed:

- The aggregates were sectioned to 10 μm using a Leica CM3050S cryostat.
- Sections were transferred to Superfrost Plus slides, which were washed with PBS to remove the OCT compound.
- The slides were washed with PBST (0.1% Triton X-100 in PBS) and sections were circled with a Pap pen.
- Slides were blocked for 30 min at room temp. with blocking buffer (1% bovine serum albumin and 5% normal donkey serum in PBST).
- The blocking buffer was removed and replaced with a solution of primary antibodies in the blocking buffer, and the slides were incubated overnight at 4 °C.
- The antibody solution was removed, and the slides were washed with PBST three times for 5 min.
- The slides were incubated with secondary antibody and DAPI solution in blocking buffer for 1 h at room temperature in the dark, followed by two 5 min washes with PBST and one wash with PBS.
- After staining, samples were mounted in Prolong Gold medium and covered with coverslips.
- Imaging was performed on a Leica SP5 confocal microscope.

Antibodies used are listed as follows:
**Primary antibodies:**

- OCT4- Mouse IgG (AB398736) - 1:250 Dilution
- DDX4- Mouse IgG (AB27591) - 1:500 Dilution
- DAZL- Rabbit IgG (AB215718) - 1:167 Dilution
- SOX17- Goat IgG (AB355060) - 1:500 Dilution
- ITGA6- Rat IgG (AB493634) - 1:167 Dilution (1:60 for flow cytometry).

**Secondary antibodies:**

- Mouse IgG-AF647 Donkey (AB162542) - 1:250 Dilution
- Rabbit IgG-AF488 Donkey (AB2534104) - 1:500 Dilution
- Goat IgG-AF568 Donkey (AB2534104) - 1:500 Dilution

## Quantification and statistical analysis

All statistical analysis and specific quantification can be found in the figure legends and methods sections. All graphing was performed using GraphPad unless otherwise noted. For Fig. 2B, $n = 3$ replicates were independently seeded for induction from a single hiPSC cell line. Fig. 2C, $n = 6$ replicates were independently seeded for induction from three independent hiPSC cell lines, two per line. Results from each cell line were averaged for the duplicates and graphed as biological replicates. For Fig. EV1, $n = 3$ replicates were independently seeded for induction from a single hiPSC cell line. For Fig. EV2, $n = 3$ replicates were generated from a single hiPSC cell line by pooling $n = 9$ independent floating aggregates.

## Data availability

All data needed to evaluate the conclusions in the paper are present in the paper and supplementary tables and figures or as source data

files. Data analysis code can be found at: https://github.com/programmablebio/egg. Raw data for microscopy images in Fig. 4 as well as flow cytometry data for Fig. 3 can be found on BioStudies Repository S-BSST1527. Raw and processed sequencing data are deposited to GEO and will be publicly available upon publication. Barcode sequencing from Fig. 3 can be found at GSE273748, RNA sequencing from Fig. 5 can be found at GSE273465, and Methyl-seq from Fig. 6 can be found at GSE273772.

The source data of this paper are collected in the following database record: biostudies:S-SCDT-10_1038-S44319-025-00371-2.

# Peer review information

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

## Acknowledgements

This work was funded by the Synthetic Biology Platform at the Wyss Institute for Biologically Inspired Engineering and institutional startup funds at Duke University. We thank Songlei Liu, Sabrina Koseki, Kalyan Palepu, Garyk Brixi, Suhaas Bhat, Emma Tysinger, and Teodora Stan for technical assistance in the research. This work was additionally funded by the Gameto Sponsored Research Agreement at Harvard University and the Colossal Sponsored Research Agreement at Harvard University. MPS was supported by the

National Science Foundation Graduate Research Fellowship and NICHD F31 Fellowship (F31HD108898-01A1).

## Author contributions

**Merrick Pierson Smela**: Formal analysis; Validation; Investigation; Visualization; Methodology; Writing—original draft; Writing—review and editing. **Christian C Kramme**: Conceptualization; Data curation; Formal analysis; Funding acquisition; Investigation; Methodology; Writing—original draft; Writing—review and editing. **Patrick R J Fortuna**: Investigation; Writing—review and editing. **Bennett Wolf**: Investigation. **Shrey Goel**: Investigation; Formal analysis; Visualization. **Jessica Adams**: Investigation. **Carl Ma**: Investigation. **Sergiy Velychko**: Investigation. **Ursula Widocki**: Investigation. **Venkata Srikar Kavirayuni**: Software; Investigation. **Tianlai Chen**: Investigation. **Sophia Vincoff**: Investigation. **Edward Dong**: Investigation. **Richie E Kohman**: Supervision. **Mutsumi Kobayashi**: Investigation. **Toshi Shioda**: Supervision; Writing—review and editing. **George M Church**: Conceptualization; Supervision; Writing—review and editing. **Pranam Chatterjee**: Conceptualization; Supervision; Investigation; Methodology; Visualization; Validation; Formal analysis; Funding Acquisition; Writing—original draft; Writing—review and editing.

Source data underlying figure panels in this paper may have individual authorship assigned. Where available, figure panel/source data authorship is listed in the following database record: biostudies:S-SCDT-10_1038-S44319-025-00371-2.

## Disclosure and competing interests statement

PC, CCK, MPS, and GC are listed as inventors for the International Patent Application No.: PCT/US2023/065143, entitled: "Methods and Compositions for Producing Primordial Germ Cell-Like Cells," and International Patent Application No.: PCT/US2023/065145, entitled: "Methods and Compositions for Producing Oogonia-Like Cells." PRJF declares paid consultancy for Gameto Inc. CCK is currently the Chief Scientific Officer of Gameto, Inc. MPS is currently the Chief Scientific Officer of Ovelle Inc. GMC serves on the scientific advisory board of Gameto, Inc., Colossal Biosciences, and GCTx. PC is a co-founder, scientific advisor to Gameto, Inc.

# Expanded View Figures

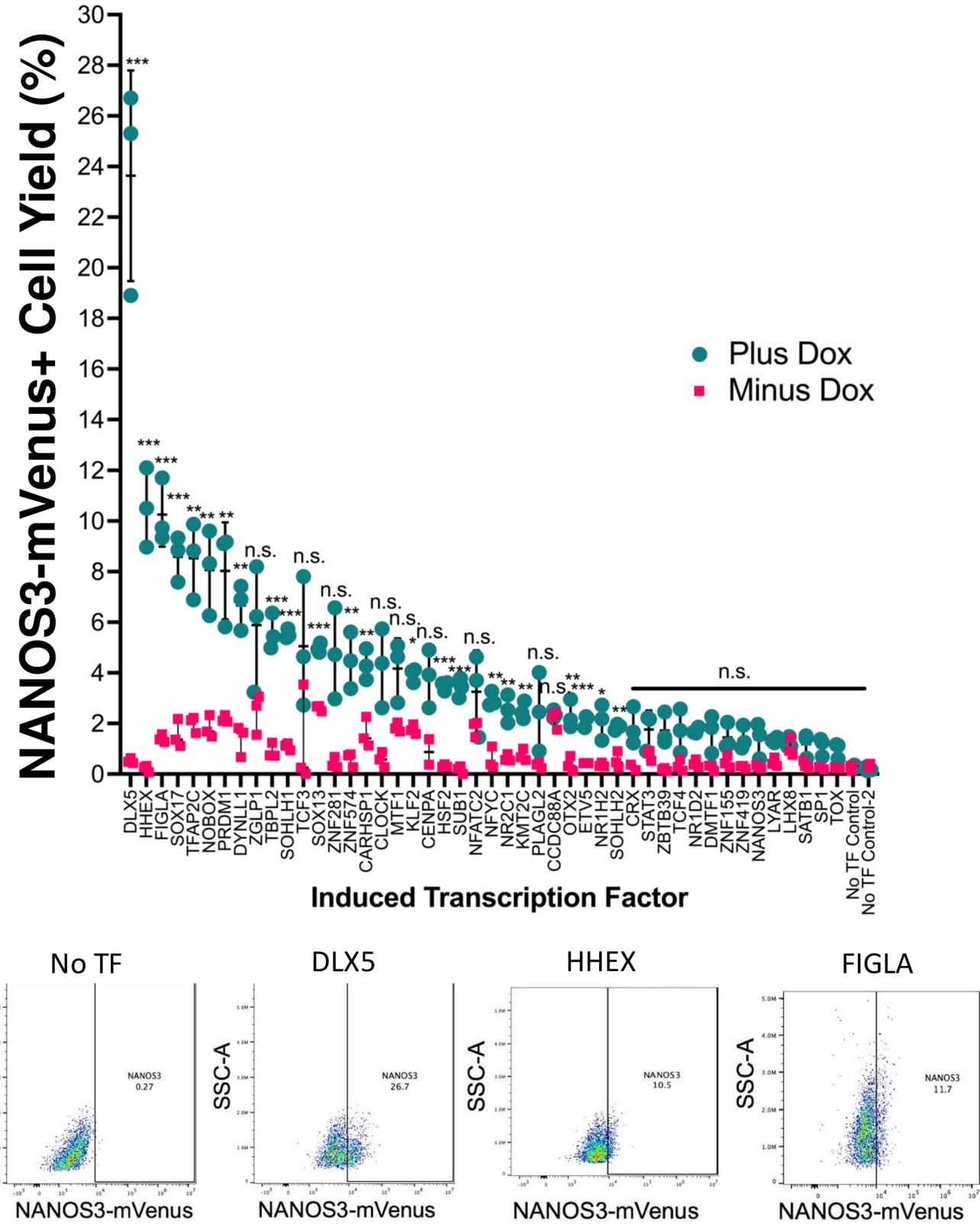

◀ **Figure EV1.  Single TF-overexpression screening for NANOS3+ cell yield, Related to Fig. 2.**

Results for NANOS3-T2A-mVenus flow cytometry for triplicate induction conditions in the plus (blue) and minus (pink) doxycycline induction condition for each TF. Data were plotted as a percent of NANOS3-T2A-mVenus+ cells on the y-axis for each TF. Individual dots represent individual wells of the induction condition, seeded separately from a single hiPSC line. The horizontal black line represents the mean of induction replicates. Statistical significance was determined by multiple *T*-test comparison between the plus and minus dox condition for each TF, with a *p* value <0.05 considered as significant. The FDR correction was utilized for multiple hypothesis testing. ***$p$ < 0.001, **$P$ < −0.01, *$p$ < 0.05. Representative flow cytometry plots are shown for the No-TF control condition and the three highest yielding conditions: DLX5, HHEX, and FIGLA.

# A

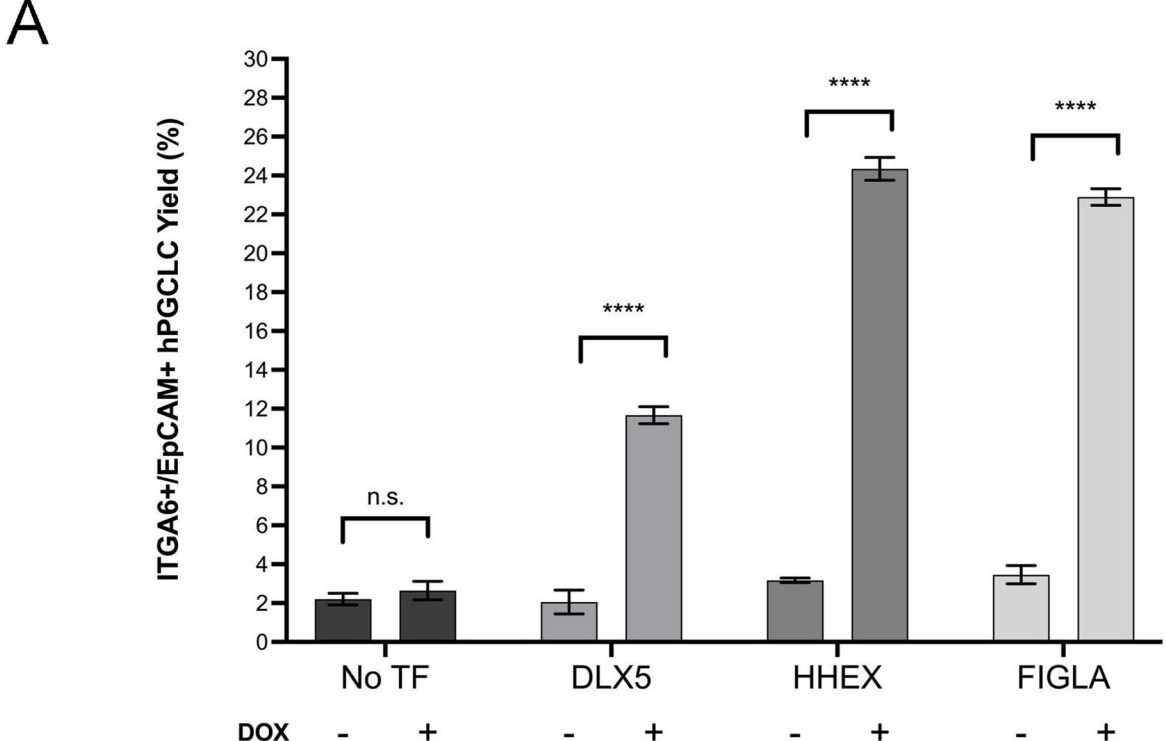

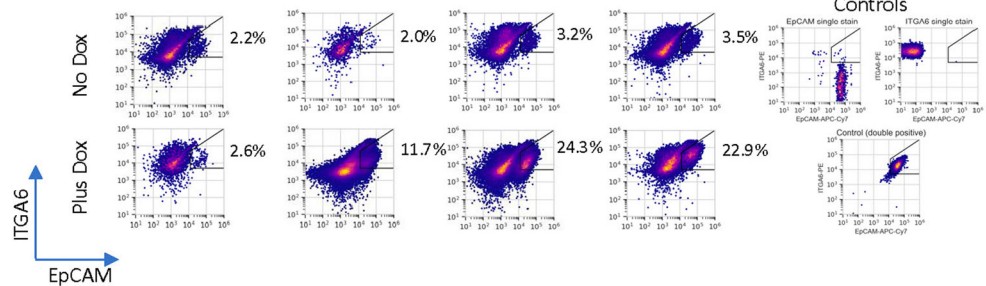

# B

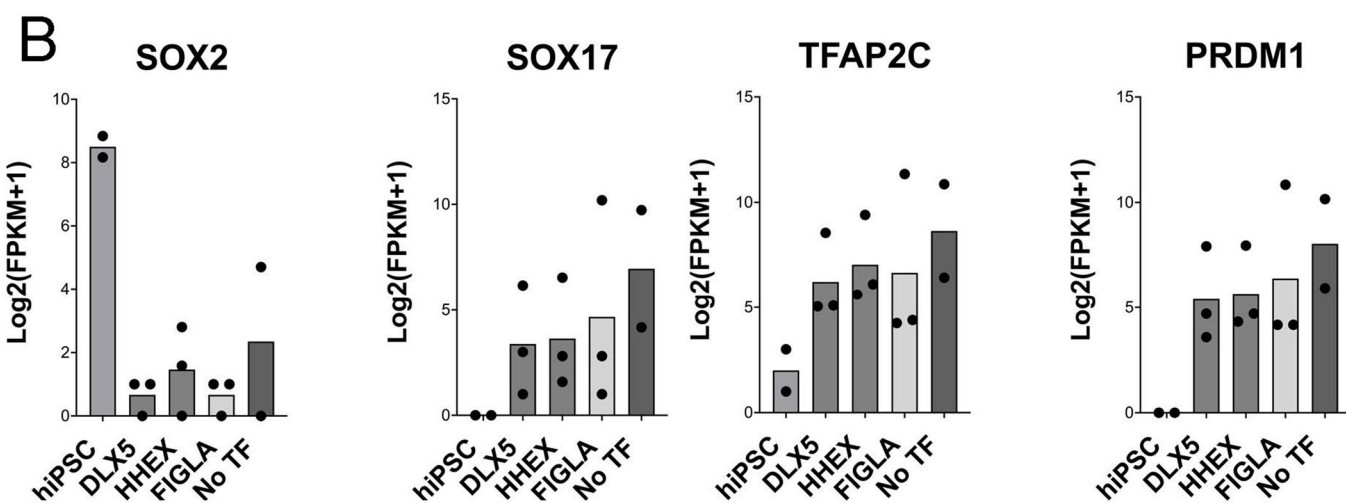

**Figure EV2.   *DLX5*, *HHEX*, and *FIGLA* overexpression significantly improves hPGCLC yield, Related to Fig. 2.**

(A) Generation of hPGCLCs in $N = 3$ replicates of $n = 9$ pooled embryoid bodies per condition. hPGCLCs were induced according to the protocol from Sasaki et al, 2015 and analyzed via flow cytometry for the markers EpCAM and ITGA6. hPGCLCs were induced in the presence and absence of doxycycline in lines harboring inducible vectors for *DLX5*, *HHEX*, and *FIGLA* and a no TF control. Statistical analysis was performed via an unpaired two-tailed *t*-test between doxycycline and no doxycycline conditions for each condition. ****$p < 0.0001$. Representative flow cytometry graphs are shown for EpCAM/ITGA6 on live singlets for the no dox and plus dox conditions as well as single stain controls. (B) Expression of known hPGCLC marker genes (*SOX17*, *TFAP2C*, *PRDM1*) and hiPSC marker gene (*SOX2*), in NANOS3+ sorted cells isolated from monolayer induction in the no TF control, *DLX5*, *HHEX*, and *FIGLA* overexpression conditions. RNA abundance was measured via RNA-seq for $N = 2$ or 3 independent replicates and plotted via Log2 normalization of FPKM + 1 values against an hiPSC control.

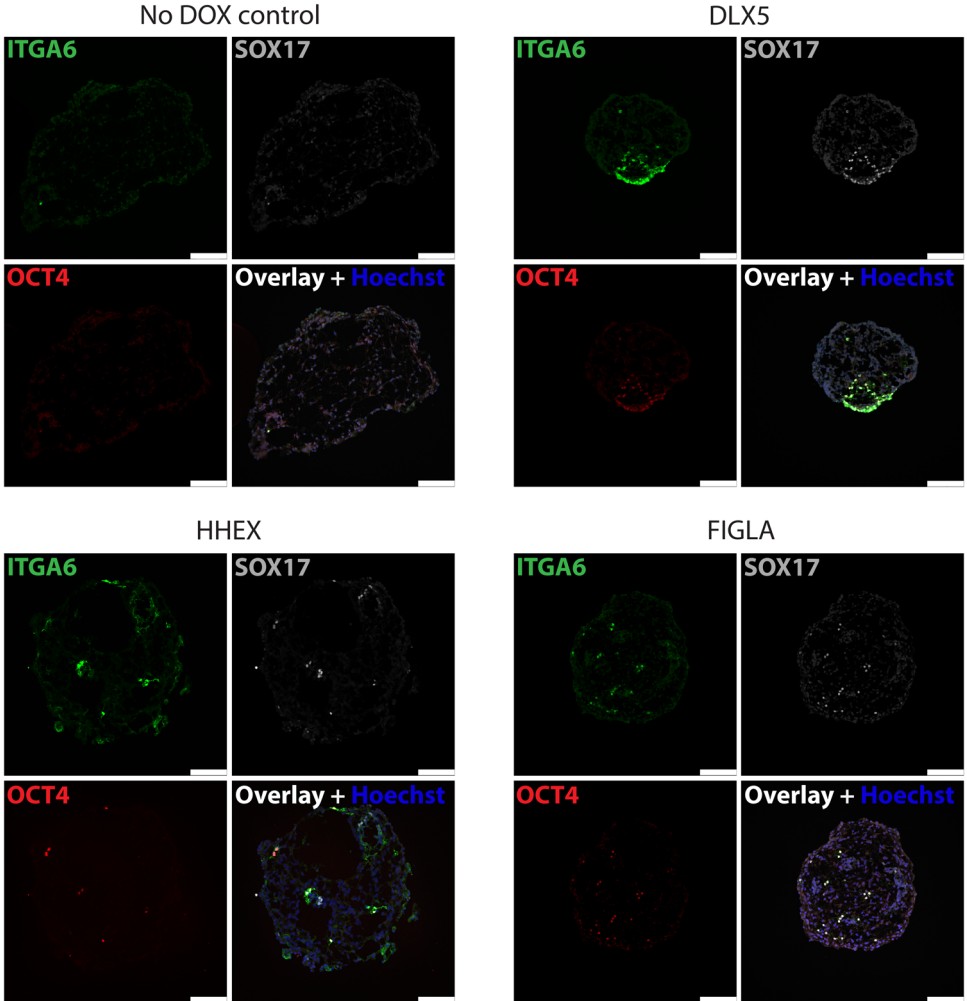

**Figure EV3.  TF-derived hPGCLCs display canonical protein expression in floating aggregate differentiation, Related to Fig. 2.**

hPGCLCs were induced according to the protocol from Sasaki et al, 2015 and analyzed via immunofluorescence after fixation and cryosectioning. hPGCLC markers ITGA6 (green), SOX17 (gray), and OCT4 (red) are visualized alongside Hoechst staining (blue). A no doxycycline control and embryoid bodies from *DLX5*, *HHEX*, and *FIGLA* inductions are shown at day 4 of differentiation. Scale bars are 100 μm.

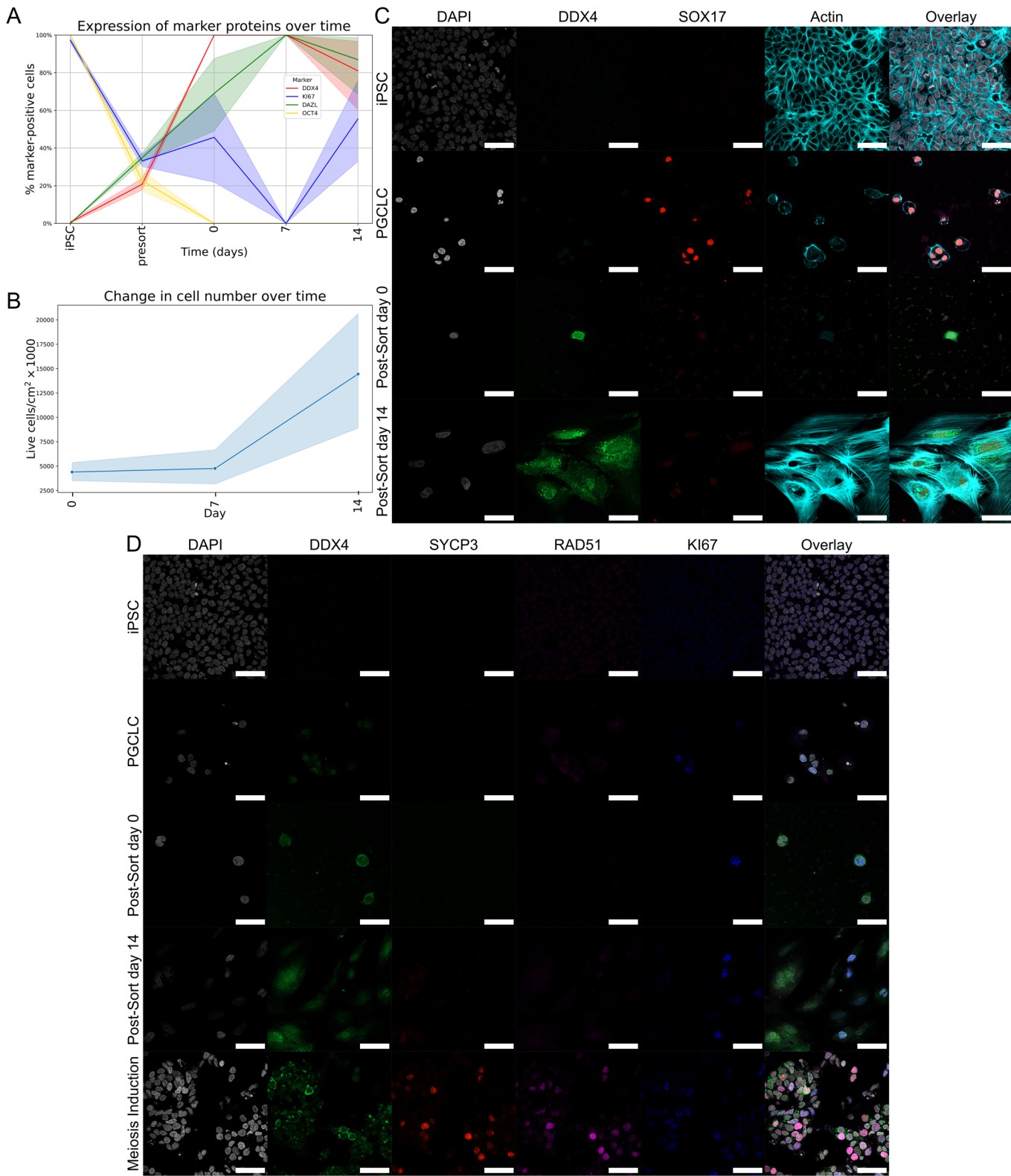

**Figure EV4.   Staining of iOLCs for PGC and meiosis marker proteins. Related to Fig. 3.**

(A) Percentage of marker-positive cells at different timepoints of iOLC induction and culture post-sorting. $N = 3$ biological replicates (F2, F3, PGP1) with at least two images per cell line. (B) The total number of cells over culture post-sorting. (C) Staining of hiPSCs, hPGCLCs, and sorted iOLCs (day 0 and day 14 post-sort) for DDX4 (green), SOX17 (red), and actin (cyan). Scale bars are 50 μm. (D) Staining for DDX4 (green), SYCP3 (red), RAD51 (magenta), and KI67 (blue). As a positive control for meiotic markers, staining was performed on cells differentiated using the meiosis induction protocol described in Pierson Smela et al 2024. Scale bars are 50 μm.

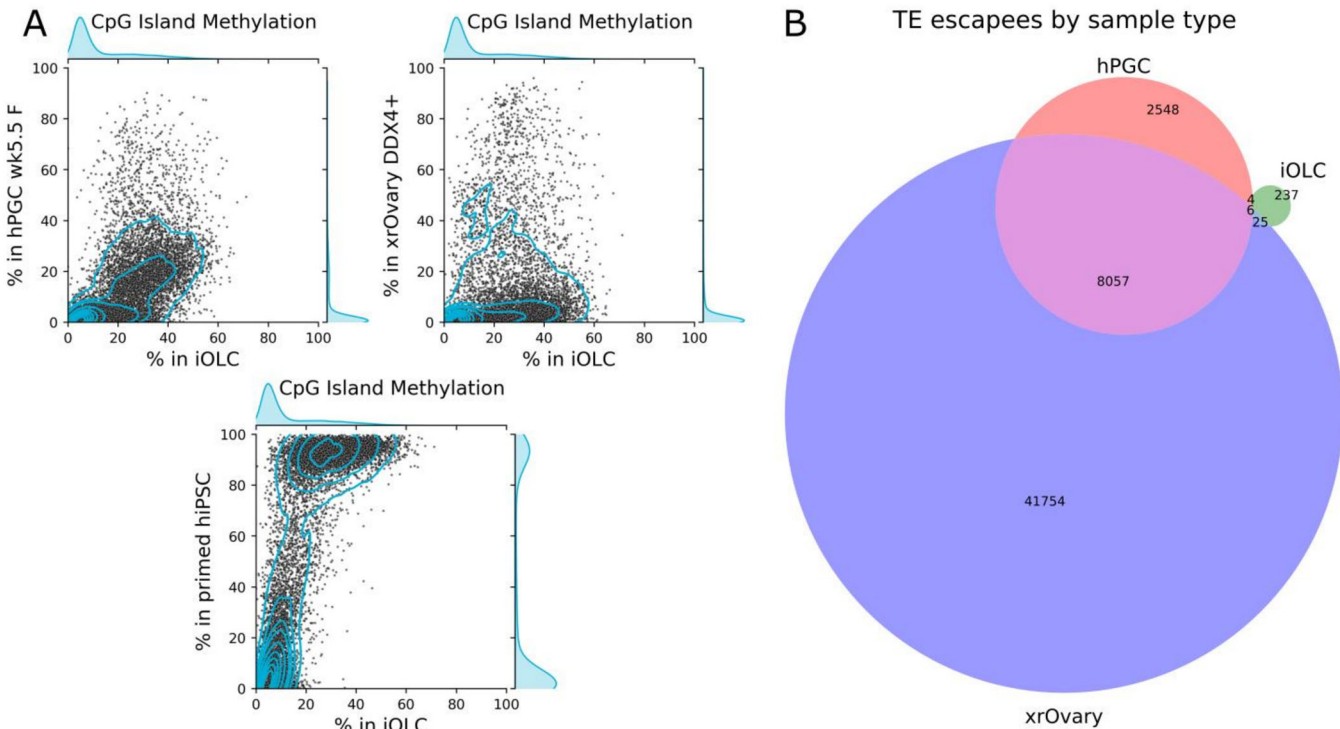

**Figure EV5.** **Additional DNA methylation analysis of iOLCs, hPGCs, and xrOvary cells. Related to Fig. 6.**

(A) CpG island methylation in primed hiPSCs, iOLCs (induced using D5 TFs and the improved protocol), week 5.5 female hPGCs, and xrOvary day 120 DDX4+ cells. (B) Overlap of TEs escaping demethylation (defined as >80% average methylation) in iOLCs, week 5.5 female hPGCs, and xrOvary day 120 DDX4+ cells.

