## [Peer Review File · EMBO Reports]

Rapid Human Oogonia-like Cell Specification via Transcription Factor-Directed Differentiation

Merrick Pierson Smela, Christian Kramme, Patrick Fortuna, Bennett Wolf, Shrey Goel, Jessica Adams, Zijian(Carl) Ma, Sergiy Velychko, Ursula Widocki, Venkata Srikar Kavirayuni, Tianlai Chen, Sophia Vincoff, Edward Dong, Richie Kohman, Mutsumi Kobayashi, Toshihiro Shioda, George Church, and Pranam Chatterjee

Corresponding author: Christian Kramme (christian@gametogen.com)

Review Timeline:

Submission Date:	2nd May 24
Editorial Decision:	3rd May 24
Revision Received:	22nd Jul 24
Editorial Decision:	14th Oct 24
Revision Received:	1st Nov 24
Editorial Decision:	4th Dec 24
Revision Received:	18th Dec 24
Accepted:	8th Jan 25

Editor: Achim Breiling

Transaction Report:

(Note: This manuscript was transferred to EMBO reports following peer review at The EMBO Journal. With the exception of the correction of typographical or spelling errors that could be a source of ambiguity, letters and reports are not edited. Depending on transfer agreements, referee reports obtained elsewhere may or may not be included in this compilation. Referee reports are anonymous unless the Referee chooses to sign their reports.)

Dear Dr. Kramme,

Thank you for transferring your manuscript to EMBO reports. I now went through the manuscript, the referee reports from The EMBO Journal (attached again below) and your provisional p-b-p-response (revision plan). The referees have several concerns and suggestions to improve the manuscript, or to strengthen the data and the conclusions drawn.

Given the constructive referee comments, I would like to invite you to revise your manuscript with the understanding that all concerns of the referees must be addressed in the revised manuscript or in a detailed point-by-point response, as indicated in your revision plan. Acceptance of your manuscript will depend on a positive outcome of another round of review at EMBO reports, using the same referees.

EMBO reports emphasizes novel functional over detailed mechanistic insight. Thus, we will not require addressing points regarding more mechanism experimentally. However, it will be necessary that during revision you address all points questioning the main conclusions of the study, and all technical concerns, or points regarding the experimental designs, model systems used, or data presentation.

1) a .docx formatted version of the final manuscript text (including legends for main figures, EV figures and tables), but without the figures included. Please make sure that changes are highlighted to be clearly visible. Figure legends should be compiled at the end of the manuscript text.

2) individual production quality figure files as .eps, .tif, .jpg (one file per figure), of main figures and EV figures. Please upload these as separate, individual files upon re-submission. Please make sure that all figure panels are called out separately and sequentially in the manuscript text

For more details please refer to our guide to authors:

See also our guide for figure preparation:

Moreover, please consult our guidelines for figure legend preparation:

4) a complete author checklist, which you can download from our author guidelines

(<https://www.embopress.org/page/journal/14693178/authorguide>). Please insert page numbers in the checklist to indicate where the requested information can be found in the manuscript. The completed author checklist will also be part of the RPF.

5) that primary datasets produced in this study (e.g. RNA-seq, ChIP-seq and array data) are deposited in an appropriate public database. This is now mandatory (like the COI statement). If no primary datasets have been deposited in any database, please state this in this section (e.g. 'No primary datasets have been generated and deposited').

The accession numbers and database should be listed in a formal "Data Availability " section (placed after Materials & Methods) that follows the model below. Please note that the Data Availability Section is restricted to new primary data that are part of this study.

Data availability

8) Regarding data quantification and statistics, please make sure that the number "n" for how many independent experiments were performed, their nature (biological versus technical replicates), the bars and error bars (e.g. SEM, SD) and the test used to calculate p-values is indicated in the respective figure legends (also for potential EV figures and all those in the final Appendix). Please also check that all the p-values are explained in the legend, and that these fit to those shown in the figure. Please provide statistical testing where applicable. Please avoid the phrase 'independent experiment', but clearly state if these were biological or technical replicates. Please also indicate (e.g. with n.s.) if testing was performed, but the differences are not significant. In case n=2, please show the data as separate datapoints without error bars and statistics.

See also:

<http://www.embopress.org/page/journal/14693178/authorguide#statisticalanalysis>

9) Please note our reference format:

10) We updated our journal's competing interests policy in January 2022 and request authors to consider both actual and perceived competing interests. Please review the policy <https://www.embopress.org/competing-interests> and add a statement declaring your competing interests. Please name that section 'Disclosure and Competing Interests Statement' and add it after the author contributions section.

11) Please provide a title with not more than 100 characters (including spaces), and abstract (in present tense) with not more than 175 words, add up to five keywords to the manuscript and order the sections like this using these names:
Title page - Abstract - Keywords - Introduction - Results - Discussion - Methods - Data availability section (DAS) - Acknowledgements - Disclosure and Competing Interests Statement - References - Figure legends - Expanded View Figure legends

12) Please make sure that all the funding information is also entered into the online submission system and is complete and similar to the one in the manuscript text file (in the Acknowledgements).

13) We now use CRediT to specify the contributions of each author in the journal submission system. CRediT replaces the author contribution section. Please use the free text box to provide more detailed descriptions. Thus, please do not provide your final manuscript text file with an author contributions section. See also guide to authors:
<https://www.embopress.org/page/journal/14693178/authorguide#authorshipguidelines>

14) We would encourage you to use 'Structured Methods', our new Materials and Methods format. According to this format, the Materials and Methods section should include a Reagents and Tools Table (listing key reagents, experimental models, software and relevant equipment and including their sources and relevant identifiers) followed by a Methods and Protocols section in which we encourage the authors to describe their methods using a step-by-step protocol format with bullet points, to facilitate the adoption of the methodologies across labs. More information on how to adhere to this format as well as downloadable templates (.doc or .xls) for the Reagents and Tools Table can be found in our author guidelines (section 'Structured Methods'):

I look forward to seeing a revised version of your manuscript when it is ready. Please let me know if you have questions or comments regarding the revision.

Kind regards,

Achim

Referee #1:

This manuscript, written by the Kramme Group, describes the induction of oogonium-like cells (iOLCs) from human induced pluripotent stem cells (hiPSCs) through the overexpression of transcription factors relevant to germ cell development. The authors first identified transcription factors through in silico screenings and then overexpressed them in DDX2-tdTomato and/or NANOS3-mVenus reporter hiPSCs. Based on the expression of the reporter iPSCs, the authors determined five transcription factors, along with a chemical inhibitor of DNMT1, that can induce iOLCs. iOLCs shared similarities in gene expression profiles and DNA methylation status with either human primordial germ cell-like cells (hPGCLCs) cultured in xenogenetic reconstituted ovary (xrOvary) or fetal oogonia.

Although this paper is potentially interesting in terms of a method for screening transcription factors involved in cell differentiation, there are many structural flaws as a manuscript that claims induction of iOLCs from hiPSCs. Unfortunately, the quality of results, including images and gene expression comparisons, is low, thereby making the conclusion ambiguous. As described below, to convince faithful induction of iOLCs from hiPSCs, there are several critical points to be clarified by the authors.

1. Definition of iOLCs is not convincing: the authors claimed that the cells induced by 5 factors (or 3 factors) are similar to oogonia with the sentence "Based on DDX4 expression and subsequent analyses showing similarities between our DDX4+ cells and oogonia, we term these induced oogonia-like cells". However, this claim is not sufficiently supported by evidence in a manner that readers can understand. The authors should have compared gene expression profiles of iOLCs with those of human germ cells in vivo and in vitro at various stages. Although a sort of this analysis is shown in Figure 6, it is quite hard to interpret and not comprehensive. The authors should perform a comprehensive comparison of the transcriptome using appropriate methods such as clustering analysis and UMAP. Otherwise, it cannot be determined whether the definition of iOLCs is reasonable.

2. Although the authors created dual reporter cell lines, there are no results from analyses using both of them. Since these reporters represent early and late germ cell development, the authors should investigate the impacts of transcription factors on the induction of germ cell lineage in hiPSCs. Simultaneously using dual reporters would provide more informative and convincing results.

3. It is not convincing that the expression level of the reporters, especially NANOS3-mVenus, is sufficiently upregulated. The upregulation of N3V seems subtle (Supplementary Figure 4), compared to PGCLCs (Irie et al., 2015, Cell). The authors should

compare the levels with PGCLCs for a more thorough assessment.

4. The author claim that iOLCs can expand in culture (Figure 3A and B). However, there is no data to support this claim. At least, the authors should demonstrate an increase in the number of iOLCs in the culture system by counting cells or conducting a proliferation assay such as BrdU incorporation.

5. Regarding iOLC expansion, the authors claim that iOLCs showed "a multi-nucleated state reminiscent of the cell state of oogonia." However, the authors should carefully observe this state with appropriate images. Firstly, the image provided is of too low magnification to interpret effectively. Secondly, oogonia do not typically exhibit a multinucleated state; rather, they display well-separated nuclei within each cytoplasm connected through intercellular bridges, which seems different from what the authors observed. Lastly, large cytoplasm with multiple nuclei is a typical phenotype of aged cells. The authors should validate these possibilities to arrive at a more convincing conclusion.

6. The authors claim that the addition of a DNMT1 inhibitor improved the DNA methylation status in iOLCs. I cannot quite understand the significance of this experiment. Is this merely a consequence of DNMT1 inhibition and too artificial? While I understand that the inhibition shows a similar demethylation status of iOLCs to human oogonia, the PCA plots (Figure 5 C and D) are widely scattered. Additionally, the colors of dots in the figures are ambiguous.

Although minor points.

- The author should display the same number of plots in each FACS panel. Otherwise, it would be hard to interpret.
- The image in Supplementary Figure 6 should be presented at a higher magnification to aid interpretation.
- FACS analysis should be shown for Figure 4D. Using the double reporter would be better.

Referee #2:

In this paper the authors used PageRank graph theory algorithm and identified 36 transcriptional factors (TFs) that are critical regulators of oogenesis. They included 10 other control TFs that are defined to have a role in late-stage oogenesis and performed a broad overexpression of these 46 TFs to screen hiPSC cell line that would be DDX4+ cells expressing tdTomato in 4days feeder-free monolayer culture conditions.

However, most of the 46 TFs are expressed in the gonadal germ cells in both male and female development.

It is unclear their reasoning of expressing AR-V7 to activate male germ cell related genes, as AR is not active at this stage of germ cell development.

Overall percentage of DDX4+ cells was not greatly enriched by any single TF and combination of 3TFs resulted in only 1.2% of DDX4+ cells.

The interesting result in their paper is use of inhibitor of DNMT1, to suppress DNA methylation. This inhibitor could be useful in further understanding of changes in methylation pattern during germ cell development.

The main positive highlight of the paper is the identification of transcription factors that initiate the differentiation of hiPSC into germ cell like in 4day feeder free monolayer culture conditions. However, like to authors point out in their discussion, they do not observe significant enrichment of genes in germ cell development between their iOLC, hPGC, and xr ovary oogonia. (Fig S8. A&B)

Therefore, their claim that they have oogonia like cells is over-interpreted.

Fig.5A The methylation in iOLC-DNM1i cell is close to hPGCLC and not in vivo hPGC 7-9wk timepoint.

Fig 6. Heatmap showing DGE of iOLC is distinct from 19 wk old Fetal Ovary.

Supp Fig 5. Repeat of the two graph's.

Referee #3:

This study identified transcription factors (TFs) whose combinatorial overexpression drives DDX4+-induced oogonia-like cell formation and three TFs whose individual overexpression enhances hPGCLC formation from hiPSCs. They have identified these TFs through analyses of available RNA-seq data and then overexpressed them in human iPSCs, testing their ability to produce DDX4+ and NANOS3+ cells after four days in culture. There are some questions arising from subsequent experiments that need to be addressed, outlined below:

1. More quantitative analysis of cultures is needed to understand better the method's efficiency and reproducibility per step and per final TF. Zoom-out view of cultures and staining for several key markers and assessment of cells that express overlapping key markers as readout necessary.
2. How will the copy number of the genetic circuit affect the outcome? How multiple Tfs are chosen -did authors use combinatorial antibiotic selection? If not, how do they make sure about the success? A discussion would be necessary. When combinatorial screening is done, are there different combinations of TFs and copy numbers used and if silencing is an issue?
3. The paper is very dense and would be challenging for less familiar readers to follow the experiments and outcomes. Authors need to organize the paper and provide schematics per step/experiment to show how the experiment is done. Media used, time of delivery of TF per engineered line plus growth factor supplementation per step. For example, how single Tf and combinatorial Tfs are used to make oogonia-like cells and hPGCLC. Schematics can help the readers follow better.

4. A better discussion of available methods in the introduction is necessary. Additionally, how this method is compared against those for efficiency of the outcome data to compare the method or benchmark against those methods helps to understand the outcome of this study better.
5. The authors have included a small molecule inhibitor of DNA methyltransferase (DNMT1) and concluded that their improved protocol was able to erase DNA methylation except in the majority of the transposable elements (TEs) regions, where their methylation ensures that they remain suppressed. The authors have shown that 280 TEs retained >80% methylation, and 8657 TEs retained >60% average methylation. However, they should demonstrate how the activated (demethylated) TEs affect the whole genome and whether this set of activated TEs exhibits deleterious effects. This becomes even more important since the authors have shown that this set of escapee TEs shows little overlap with escapee TEs observed in vivo.
6. Due to different sequencing techniques, the authors could not compare their iOLCs and in vivo germ cells. However, the manuscript would benefit from single-cell RNA seq data to determine the in vivo developmental stage at which the iOLCs match and to compare the model to in vivo oogenesis.
7. Figure 6 is important but not clear enough. How should the similarities and differences be interpreted? What about other genes? We also need a whole genome analysis to understand off-target effects and aberrant signatures.
8. Authors are checking DEG and GO enrichment between their population of cells and other populations. Cell fate assessment methods for more computationally rigorous comparisons between populations exist and could be applied to increase confidence about the stage and state of these cells in relation to published datasets. e.g., machine learning algorithms, can be used to better rank the cell fate/ state using whole genome analysis.
9. Cell phenotypic analysis is quite narrow, making it very hard to know whether what is produced is indeed the cell type of interest.
10. DLX5 is a known amnion marker and HHEX is an endoderm marker, they should make some statement about the role of these embryonic tissues in contributing to the germline.

Referee #1:

This manuscript, written by the Kramme Group, describes the induction of oogonium-like cells (iOLCs) from human induced pluripotent stem cells (hiPSCs) through the overexpression of transcription factors relevant to germ cell development. The authors first identified transcription factors through in silico screenings and then overexpressed them in DDX2-tdTomato and/or NANOS3-mVenus reporter hiPSCs. Based on the expression of the reporter iPSCs, the authors determined five transcription factors, along with a chemical inhibitor of DNMT1, that can induce iOLCs. iOLCs shared similarities in gene expression profiles and DNA methylation status with either human primordial germ cell-like cells (hPGCLCs) cultured in xenogenetic reconstituted ovary (xrOvary) or fetal oogonia.

Although this paper is potentially interesting in terms of a method for screening transcription factors involved in cell differentiation, there are many structural flaws as a manuscript that claims induction of iOLCs from hiPSCs. Unfortunately, the quality of results, including images and gene expression comparisons, is low, thereby making the conclusion ambiguous. As described below, to convince faithful induction of iOLCs from hiPSCs, there are several critical points to be clarified by the authors.

1. Definition of iOLCs is not convincing: the authors claimed that the cells induced by 5 factors (or 3 factors) are similar to oogonia with the sentence "Based on DDX4 expression and subsequent analyses showing similarities between our DDX4+ cells and oogonia, we term these induced oogonia-like cells". However, this claim is not sufficiently supported by evidence in a manner that readers can understand. The authors should have compared gene expression profiles of iOLCs with those of human germ cells in vivo and in vitro at various stages. Although a sort of this analysis is shown in Figure 6, it is quite hard to interpret and not comprehensive. The authors should perform a comprehensive comparison of the transcriptome using appropriate methods such as clustering analysis and UMAP. Otherwise, it cannot be determined whether the definition of iOLCs is reasonable.

We thank the reviewer for this constructive criticism in terms of the presentation of transcriptomic data in the manuscript. We have provided a comprehensive revision of the manuscript to both provide new analysis of the oogonia-like phenotypic characterization as well as a re-ordering of the results section to provide better clarity and presentation of results. In the revised manuscript, we now utilize Figure 2 and Figure 3 to identify transcription factor combinations that improve the yield of DDX4+ cells based on flow cytometry, identifying the core 5 transcription factors used in our manuscript. In Figure 4 and Figure S7, we then perform immunofluorescence analysis of the induced cells in order to provide a more comprehensive assessment of the phenotype that supports the designation of the DDX4+ cells as induced oogonia-like cells (iOLCs). Specifically, we test for pluripotency (OCT4), PGCLC (SOX17), proliferation (KI67), oogonia (DDX4, DAZL), and meiosis (SYCP3, RAD51) protein expression (Figure 4). Based on the phenotype, we then designate these TF-derived cells as iOLCs in the following sentence "Based on our live imaging (Figure 4A) and immunofluorescence staining (Figure 4B), we therefore conclude

that our TF-derived DDX4+ cells are induced oogonia-like cells (iOLCs) based on expression of mitotic oogonia markers (DDX4, DAZL, KI67) and lack of earlier primordial germ cell and pluripotency markers (OCT4, SOX17) and lack of later meiotic markers (SYCP3, RAD51) and are capable of feeder-free expansion while retaining this oogonia-like identity post-sorting.” We believe that by assessing the phenotype using well-described and established marker genes for each stage of germline development, we present a more evidenced-based metric by which to designate these cells as iOLCs. We then further use Figures 5 and 6 to both expand on our phenotypic characterization and compare to other in vivo and in vitro germ cells using both transcriptomics (Figure 5, Figure S9) and epigenetics (Figure 6, Figure S8). In Figure 5, we utilized well described marker genes for germ cell characterization from both Murase et al. 2024 and Yamashiro et al. 2024 as well as markers for off-targets of the three germ layers and pluripotency. We then use unsupervised hierarchical clustering (UHC) to compare our iOLCs to other established in vivo and in vitro germ cell specimens, and identify our iOLCs as belonging to the cluster containing advanced Week 19 fetal germ cells and Day 120 DDX4+,TFAP2C- xrOvary oogonia. This cluster displays expression of key oogonia genes and generally reduced expression of PGC, pluripotency, endoderm, ectoderm, and mesoderm genes with limited expression of meiotic genes. This is then further corroborated by machine-learning based classification of the iOLCs as oogonia using the human fetal gonad reference annotations (Figure S9). In Figure 6, we likewise establish our D5 method with DNTMi shows an epigenetic phenotype most similar to fetal germ cells and advanced xrOvary samples by both average CpG methylation (Figure 6A), and PCA (Figure 6D and E). We believe that this revision, which utilizes more comprehensive IF markers, and further transcriptomic analysis provides both more robust evidence for designating these cells as iOLCs and likewise provides a more comprehensible presentation of the similarity of our cell phenotype to those described by others in the field.

2. Although the authors created dual reporter cell lines, there are no results from analyses using both of them. Since these reporters represent early and late germ cell development, the authors should investigate the impacts of transcription factors on the induction of germ cell lineage in hiPSCs. Simultaneously using dual reporters would provide more informative and convincing results.

We thank the reviewer for this suggested edit. In the revised manuscript dual reporter outcomes for DAZL/DDX4 are shown in Figure 3E. In the revised manuscript, the outcomes in Figure 2B show single TF screening is DDX4 negative, while Figure S4 shows single TF screening is NANOS3 positive in the dual reporter for NANOS3/DDX4, and therefore these are not shown on the same graph (as the cells are single positive).

3. It is not convincing that the expression level of the reporters, especially NANOS3-mVenus, is sufficiently upregulated. The upregulation of N3V seems subtle (Supplementary Figure 4), compared to PGCLCs (Irie et al., 2015, Cell). The authors should compare the levels with PGCLCs for a more thorough assessment.

We understand this limitation and believe it is an important technical point to address. In our manuscript, we utilized a NANOS3-mVenus reporter line that was indeed faint. However, as Supplementary Figure 2F shows, these NANOS3-mVenus positive cells are nearly universally EpCAM/ITGA6 dual positive, which shows they clearly overlap with established markers for PGCLCs, validating its use in the screen. The usage of this reporter was to screen in parallel a number of different TFs to identify those capable of increasing or decreasing the percentage of NANOS3+ cells, which could be readily ascertained with the reporter level expression. However, in our manuscript, we do not claim that all NANOS3+ cells from the screen in Supplementary Figure 4 are PGCLCs, as this is only one marker that was used to down-select candidate TFs for further characterization. In order to then validate these TFs as being capable of increasing bonafide PGCLC differentiation yield, we utilized canonical cell surface markers of PGCLCs, ITGA6 and EpCAM, to validate an increase in hPGCLC formation via overexpression of DLX5, HHEX, and FIGLA (the top candidates from the NANOS3 screen) as can be seen in Supplementary Figure 5A. These markers form a clear population for flow cytometry analysis, which is significantly increased in the induction condition for all 3 TFs compared to the non-induction condition in the absence of TF presence. We further corroborate these findings with gene expression analysis of key PGCLC genes SOX17, TFAP2C, and PRDM1 in Supplementary Figure 5B, as well as IF analysis of SOX17, ITGA6, and OCT4 expression in Supplementary Figure 6. Based on these findings, we conclude that induction of DLX5, HHEX, and FIGLA enhances hPGCLC specification.

4. The author claim that iOLCs can expand in culture (Figure 3A and B). However, there is no data to support this claim. At least, the authors should demonstrate an increase in the number of iOLCs in the culture system by counting cells or conducting a proliferation assay such as BrdU incorporation.

We thank the reviewer for this concern, as part of the revised manuscript we have now provided IF analysis of KI67 which can be seen in Figure 4B and further quantified in Figure S7A. We further provide a cell count measurement over time which can be seen in Figure S7B. We believe this data supports the cells as maintaining and proliferating over time post-isolation.

5. Regarding iOLC expansion, the authors claim that iOLCs showed "a multi-nucleated state reminiscent of the cell state of oogonia." However, the authors should carefully observe this state with appropriate images. Firstly, the image provided is of too low magnification to interpret effectively. Secondly, oogonia do not typically exhibit a multinucleated state; rather, they display well-separated nuclei within each cytoplasm connected through intercellular bridges, which seems different from what the authors observed. Lastly, large cytoplasm with multiple nuclei is a typical phenotype of aged cells. The authors should validate these possibilities to arrive at a more convincing conclusion.

We thank the reviewer for this important critique. We agree this statement was not well evidenced with the method of analysis originally provided and have removed this statement from the revised manuscript.

6. The authors claim that the addition of a DNMT1 inhibitor improved the DNA methylation status in iOLCs. I cannot quite understand the significance of this experiment. Is this merely a consequence of DNMT1 inhibition and too artificial? While I understand that the inhibition shows a similar demethylation status of iOLCs to human oogonia, the PCA plots (Figure 5 C and D) are widely scattered. Additionally, the colors of dots in the figures are ambiguous.

In our manuscript, we note that the original 3 TFs (D3) used to make iOLCs, LHX8, SOHLH1, and ZNF281 are able to drive DDX4+ formation (Figure 2) with additional transcriptomic (Figure 5) and protein expression features (Figure 4) of oogonia-like cells, but that these cells fail to recapitulate the epigenetic reset known of oogonia (Figure 6). As this epigenetic reset is an important feature of oogonia-like phenotype, we aimed to devise a method to recapitulate it in our system in a manner that is fast and efficient. To address this we used an approach of DNMT1 inhibition and transient naive reset of hiPSCs to better recapitulate this epigenetic blank slate state, finding improved similarity to in vivo germ cells as well as better iOLC yields (Figure 6A, C, D, E). In the updated manuscript, we revised the color of the PCA in Figure 6 to make interpretation easier. While the PCA does show a wide scatter, likely due to batch effects, we note in the results section for Figure 6C “Along principal component 1, which captured most of the variance, our iOLCs were similar to xrOvary oogonia and hPGCs (Figure 6C).” In summary, the significance of this experiment is that while our TFs are able to generate iOLCs with oogonia-like gene and protein expression profiles, due to the rapid nature of the method, they can skip the epigenetic reset which necessitates the DNTMi method to recapitulate a more in vivo-like epigenetic profile.

Although minor points.

a. The author should display the same number of plots in each FACS panel. Otherwise, it would be hard to interpret.

b. The image in Supplementary Figure 6 should be presented at a higher magnification to aid interpretation.

c. FACS analysis should be shown for Figure 4D. Using the double reporter would be better.

We thank the reviewer for these suggested minor edits, throughout the manuscript and as noted above we have shown the dual reporter outcomes, improved the IF analysis and clarity, and presented the FACS data as captured.

Referee #2:

In this paper the authors used PageRank graph theory algorithm and identified 36 transcriptional factors (TFs) that are critical regulators of oogenesis. They included 10 other control TFs that are defined to have a role in late-stage oogenesis and performed a broad overexpression of these 46 TFs to screen hiPSC cell line that would be DDX4+ cells expressing tdTomato in 4days feeder-free monolayer culture conditions.

However, most of the 46 TFs are expressed in the gonadal germ cells in both male and female development.

It is unclear their reasoning of expressing AR-V7 to activate male germ cell related genes, as AR is not active at this stage of germ cell development.

Although the reviewer is correct that the androgen receptor is not expressed in oogonia, it is expressed at a low level in pre-spermatogonia according to a recently published scRNAseq dataset of human fetal gonads (Garcia-Alonso *et al.* 2022). Regardless, our reason for examining the transcriptional effects of AR-V7 was because it appeared to promote the formation of DDX4+ cells in our barcode enrichment experiments (Figure 3A, 3B, 3C). As it turned out, AR-V7 did not have promising effects on gene expression, and in our optimized protocol we omitted AR-V7. We therefore noted this in our results and discussion sections to clarify this point.

Overall percentage of DDX4+ cells was not greatly enriched by any single TF and combination of 3TFs resulted in only 1.2% of DDX4+ cells.

Indeed the initial yield of the system using the 3 TFs LHX8, SOHLH1, and ZNF281 are capable of yielding DDX4+ cells but only at a low rate (1.2%) (Figure 2C). It should be noted that even in other systems, such as the xrOvary (Yamashiro *et al.* 2018) that the true yield of DDX4+ oogonia is quite low (0.5%-1.2% or less than ~100-500 total oogonia cells per xrOvary) after 120 days of culture, as only a fraction of input germ cells survive and develop to that advanced state in their system. In our system, we likewise show a similar percentage of DDX4+ iOLCs; however, this is formed after merely 4 days of feeder-free culture and can be effectively scaled to yield far more total cells than other methods. Furthermore, as our results show in Figure 3D the addition of 2 TFs to the D3 combo ANHX and ZGLP1 as well as naive resetting/DNMTi increases the yield of DDX4+ cells to 10-20%, a vast (~30-fold) improvement over not only our original method but also other methods in the field. We further show these findings in 3 independent hiPSC lines as well as 2 rhesus macaque iPSC lines, showing reproducibility and consistency of the method. Therefore, our improved method shows fast, efficient, and robust DDX4+ cell formation.

The interesting result in their paper is use of inhibitor of DNMT1, to suppress DNA methylation. This inhibitor could be useful in further understanding of changes in methylation pattern during germ cell development.

The main positive highlight of the paper is the identification of transcription factors that initiate the differentiation of hiPSC into germ cell like in 4day feeder free monolayer culture conditions. However, like to authors point out in their discussion, they do not observe significant enrichment

of genes in germ cell development between their iOLC, hPGC, and xr ovary oogonia. (Fig S8. A&B)

Therefore, their claim that they have oogonia like cells is over-interpreted.

We thank the reviewer for this critique. In our revised manuscript we were able to incorporate new methods and new external datasets to significantly improve our batch effect correction. In Figure 5, we indeed show our iOLCs display clear expression of oogonia marker genes, and in the revised manuscript we have provided updated graphs that make interpretation easier and also perform a clustering (Figure 5) and classification analysis (Figure S9). As can be seen our iOLCs readily cluster with Week 19 fetal germ cells as well as xrOvary Day 120 DDX4+, TFAP2C- cells from other studies and classify as oogonia. These iOLCs likewise show broad expression of oogonia marker genes, and lack of expression of earlier germ cell, pluripotency, and off-target endoderm, mesoderm, and ectoderm genes.

Fig.5A The methylation in iOLC-DNM1i cell is close to hPGCLC and not in vivo hPGC 7-9wk timepoint.

We thank the reviewer for this note. Yes this is correct, the iOLC-DNMT1i condition uses the 3 TFs without DNMTi (the (-) means minus) and indeed shows that while we can obtain iOLCs with transcriptome similarity to oogonia (Figure 5), that the epigenetic state does not recapitulate the known reset of oogonia, and looks more similar to hPGCLCs (Figure 6A). Therefore, we apply the DNMTi and reset strategies (samples iOLC D3, iOLC D5, and iOLC Reset D5) which show a far more robust epigenetic profile similar to hPGCs and xrOvary Day 120 samples (Figure 6).

Fig 6. Heatmap showing DGE of iOLC is distinct from 19 wk old Fetal Ovary.

We thank the reviewer for this note. In the updated Figure 5 heatmap, we demonstrate with marker gene expression and UHC broad similarity of our iOLCs to the Week 19 Fetal ovary cells.

Supp Fig 5. Repeat of the two graph's.

We thank the reviewer for identifying this error, we have corrected this figure in the revised manuscript to show gene expression of SOX17, TFAP2C, PRDM1, and SOX2.

Referee #3:

This study identified transcription factors (TFs) whose combinatorial overexpression drives DDX4+-induced oogonia-like cell formation and three TFs whose individual overexpression enhances hPGCLC formation from hiPSCs. They have identified these TFs through analyses of available RNA-seq data and then overexpressed them in human iPSCs, testing their ability to

produce DDX4+ and NANOS3+ cells after four days in culture. There are some questions arising from subsequent experiments that need to be addressed, outlined below:

1. More quantitative analysis of cultures is needed to understand better the method's efficiency and reproducibility per step and per final TF. Zoom-out view of cultures and staining for several key markers and assessment of cells that express overlapping key markers as readout necessary.

We thank the reviewer for this suggestion. In the updated manuscript in Figure 4 and Figure S7, we have provided a more comprehensive IF analysis including markers of oogonia (DDX4, DAZL), early PGC (SOX17), pluripotency (OCT4), proliferation (KI67), and meiosis (SYCP3 and RAD51). In Figure S7A we likewise perform quantification of marker expression in the IF. We provide both magnified and zoom-out views as well as image quantification to better aid interpretation of results and overlap.

2. How will the copy number of the genetic circuit affect the outcome? How multiple Tfs are chosen -did authors use combinatorial antibiotic selection? If not, how do they make sure about the success? A discussion would be necessary. When combinatorial screening is done, are there different combinations of TFs and copy numbers used and if silencing is an issue?

In this manuscript, the method utilized was high copy number integration and polyclonal screening. A single antibiotic selection was utilized to select for the transposons and inductions were performed in polyclonal pools. As can be seen in Figure 5, the TFs are clearly expressed in the combination cell pools, and for individual expression pools, each TF is represented (Supplementary Figure 3B). In order to overcome limitations with transposon methods, a high copy number was utilized (to prevent suppression and to ensure multiple copies of each TF) and polyclonal pools were utilized. Our manuscript does not characterize single cell cloning and the contribution of specific TF dosages to outcome, although Supplementary Figure 3C does show that these TFs are actively expressed and tuned to doxycycline concentration. In our method, we screen a large number of TFs and arrive at a combination that when integrated into multiple independent stem cell lines in both male and female samples, as well as rhesus monkey, is able to generate a consistent DDX4+ cell outcome (Figure 3D). We have provided updated materials and methods as well as discussions and limitations to better contextualize our approach in the revised manuscript.

3. The paper is very dense and would be challenging for less familiar readers to follow the experiments and outcomes. Authors need to organize the paper and provide schematics per step/experiment to show how the experiment is done. Media used, time of delivery of TF per engineered line plus growth factor supplementation per step. For example, how single Tf and combinatorial Tfs are used to make oogonia-like cells and hPGCLC. Schematics can help the readers follow better.

We thank the reviewer for this critique, in the updated manuscript we have provided revised schematic figures and M&M to make the experimental design clearer. In Figures

2A, S2A, S3A we have provided schematics of key engineering and screening experiments. We have likewise provided a comprehensive re-drafting of the manuscript's presentation to better present the methods and improve the clarity and flow of the manuscript for interpretation.

4. A better discussion of available methods in the introduction is necessary. Additionally, how this method is compared against those for efficiency of the outcome data to compare the method or benchmark against those methods helps to understand the outcome of this study better.

In the revised manuscript, we have provided more in depth discussion comparing our approach to other approaches for both the introduction and discussion sections.

5. The authors have included a small molecule inhibitor of DNA methyltransferase (DNMT1) and concluded that their improved protocol was able to erase DNA methylation except in the majority of the transposable elements (TEs) regions, where their methylation ensures that they remain suppressed. The authors have shown that 280 TEs retained >80% methylation, and 8657 TEs retained >60% average methylation. However, they should demonstrate how the activated (demethylated) TEs affect the whole genome and whether this set of activated TEs exhibits deleterious effects. This becomes even more important since the authors have shown that this set of escapee TEs shows little overlap with escapee TEs observed *in vivo*.

We thank the reviewer for this comment, but wish to provide a clarification. The reviewer's statement that our "improved protocol was able to erase DNA methylation except in the majority of the transposable elements (TEs) regions, where their methylation ensures that they remain suppressed." is incorrect, because the vast majority of TEs were demethylated in both our iOLCs, xrOvary oogonia-like cells, and *in vivo*. To put a concrete number on this, we analyzed a total of 3,152,480 TEs, and only 8657 (0.3% of the total) retained >60% methylation in our iOLCs. Similar levels of demethylation are present *in vivo*. "Escapees" is the standard term for the small fraction of TEs which retain methylation (escaping demethylation). Although our cells had a different set of escapee TEs than *in vivo*, the overall demethylation is quite similar. The reviewer's suggestion to analyze the potential deleterious effects of TEs over long-term iOLC culture is an interesting idea, but outside the scope of the current work. Still, we note that our iOLCs express germline genes such as PIWIL1, PIWIL2, and MAEL (Figure 5) which are known to inhibit TE activity by RNA-mediated silencing. We have updated the discussion section to mention the potential of deleterious TE activity and discuss silencing pathways.

6. Due to different sequencing techniques, the authors could not compare their iOLCs and *in vivo* germ cells. However, the manuscript would benefit from single-cell RNA seq data to determine the *in vivo* developmental stage at which the iOLCs match and to compare the model to *in vivo* oogenesis.

We thank the reviewer for this suggestion. We however believe that the claims of the manuscript, with updated bulk transcriptomic assessment and updated IF imaging, are sufficient methods to characterize the cell types and that while single cell sequencing would be of interest, it is not within the scope of this manuscript. In Figure 5 and Figure 6 we provide both bulk transcriptomic and epigenetic comparison to both in vivo germ cells and in vitro-derived germ cells using alternative methods.

7. Figure 6 is important but not clear enough. How should the similarities and differences be interpreted? What about other genes? We also need a whole genome analysis to understand off-target effects and aberrant signatures.

In the updated manuscript, we provide UHC (Figure 5) and cell classification analysis (Figure S9) to better visualize and represent this data as well as provide whole transcriptome analysis. We utilize key marker genes derived from well established alternative methods to generate oogonia-like cells (Yamashiro et al. 2018 and Murase et al. 2024) as well as markers of off-target and advanced germ cell types to display the dataset. UHC shows clear grouping of our iOLCs with advanced Week 19 Fetal gonad cells and Day 120 xrOvary oogonia. Our machine-learning classification likewise classifies our cells as oogonia using the Garcia-Alonso et al. 2022 Human Fetal Gonad Atlas. We believe these figures are beneficial to visualize key marker genes associated with distinct cell states and that the above methods improve the interpretation of results in the revised manuscript.

8. Authors are checking DEG and GO enrichment between their population of cells and other populations. Cell fate assessment methods for more computationally rigorous comparisons between populations exist and could be applied to increase confidence about the stage and state of these cells in relation to published datasets. e.g., machine learning algorithms, can be used to better rank the cell fate/ state using whole genome analysis.

We agree with the reviewer that additional analysis using more computationally rigorous methods were needed. As such, in the revised manuscript we provide both UHC and machine learning guided cell classification in the updated Figure 5 and Figure S9.

9. Cell phenotypic analysis is quite narrow, making it very hard to know whether what is produced is indeed the cell type of interest.

We agree with the reviewer and have sought to provide a more comprehensive phenotypic characterization of oogonia-like in our DDX4+ iOLCs. In the updated Figure 4, we provide a more comprehensive IF analysis using markers of oogonia (DDX4, DAZL), PGCs (SOX17), pluripotency (OCT4), proliferation (KI67), and meiosis (RAD51 and SYCP3) to show more concretely the DDX4+ cells are indeed oogonia-like. In updated Figure 5 and 6 we then provide more comprehensive transcriptomic and epigenetic analysis to show marker gene expression and epigenetic profile compared to reference in vivo and in vitro analogous cell types. We likewise show lack of other markers such as endoderm, ectoderm and mesoderm (Figure 5) and use cell classification algorithms to show lack of off-target

classification (Figure S9). We believe that together this updated analysis provides a more robust analysis of the phenotype presented and lends better evidence to our claim of an oogonia-like cell state.

10. DLX5 is a known amnion marker and HHEX is an endoderm marker, they should make some statement about the role of these embryonic tissues in contributing to the germline.

We thank the reviewer for pointing this out and have added this to the discussion.

Dear Dr. Kramme,

Thank you for the submission of your revised manuscript to our editorial offices. As you know, none of the three reviewers that had assessed the study at The EMBO Journal agreed to look into the revised manuscript or has submitted a report. I have therefore involved an expert arbitrator who has assessed the revision to judge if the revised manuscript adequately addresses the concerns of the referees and if the paper is suitable for publication in EMBO reports.

I have now received the report from the arbitrator, you will find below. As you will see, the arbitrator supports the publication of the study in EMBO reports. However, s/he has remaining concerns and suggestions to improve the manuscript, I ask you to address in a final revised manuscript. Please also provide a final p-b-p-response addressing these points.

- Please provide a final title with not more than 100 characters (including spaces).
 - Please provide a final abstract with not more than 175 words.
 - The Expanded View format, which will be displayed in the main HTML of the paper in a collapsible format, has replaced the Supplementary information. You can submit up to 5 images as Expanded View. Please follow the nomenclature Figure EV1, Figure EV2 etc. The figure legend for these should be included in the main manuscript document file in a section called Expanded View Figure Legends after the main Figure Legends section. Additional Supplementary material should be supplied as a single pdf file labeled Appendix. The Appendix should have page numbers and needs to include a table of content on the first page (with page numbers) and legends for all content. Please follow the nomenclature Appendix Figure Sx, Appendix Table Sx etc. throughout the text, and also label the figures and tables according to this nomenclature.
 - We now use CRediT to specify the contributions of each author in the journal submission system. CRediT replaces the author contribution section. Please use the free text box to provide more detailed descriptions. Thus, please do NOT provide your final manuscript text file with an author contributions section. See also guide to authors: <https://www.embopress.org/page/journal/14693178/authorguide#authorshippinguidelines>
 - There are author name discrepancies. It is Merrick Pierson Smela in the manuscript text file, but Merrick Smela in the submission system; and Venkata Srikar Kavirayuni in the text and Venkata Srikar in the submission system. Please check.
 - We updated our journal's competing interests policy in January 2022 and request authors to consider both actual and perceived competing interests. Please review the policy <https://www.embopress.org/competing-interests> and add a statement declaring your competing interests. Please name that section 'Disclosure and Competing Interests Statement' and add it after the author contributions section.
 - Please add up to five keywords to the manuscript and order the manuscript sections like this, using these names: Title page - Abstract - Keywords - Introduction - Results - Discussion - Methods - Data availability section - Acknowledgements - Disclosure and Competing Interests Statement - References - Figure legends - Expanded View Figure legends
 - Please remove the paragraph regarding Suppl. Figures and Tables from the manuscript text file.
 - Please make sure that the number "n" for how many independent experiments were performed, their nature (biological versus technical replicates), the bars and error bars (e.g. SEM, SD) and the test used to calculate p-values is indicated in the respective figure legends. Please also check that all the p-values are explained in the legend, and that these fit to those shown in the figure. Please provide statistical testing where applicable. Please avoid the phrase 'independent experiment', but clearly state if these were biological or technical replicates. Please also indicate (e.g. with n.s.) if testing was performed, but the differences are not significant. In case n=2, please show the data as separate datapoints without error bars and statistics. See also: <http://www.embopress.org/page/journal/14693178/authorguide#statisticalanalysis>
- If n<5, please show single datapoints for diagrams. Moreover:
- Please provide the exact p values in the legends of figures 2c; 3d.
 - Please indicate the statistical test used for data analysis in the legends of figures 1b; 3d.
 - Please note that in figure 2c; there is a mismatch between the annotated p values in the figure legend and the annotated p values in the figure file that should be corrected.
 - Please note that the box plots need to be defined in terms of minima, maxima, centre, bounds of box and whiskers, and percentile in the legends of figures 3d; 6e.
 - Please provide information related to n in the legends of figures 3c-d; 6e.
 - Please note that the measure of centre for the error bars needs to be defined in the legend of figure 3c.

- Please add scale bars of similar style and thickness to all microscopic images (main, EV and Appendix figures), using clearly visible black or white bars (depending on the background). Please place these in the lower right corner of the images themselves. Please do not write on or near the bars in the image but define the size in the respective figure legend.

- Please add that the specific URLs for GSE273748, GSE273465, GSE273772, S-BSST1527 datasets to the data availability statement.

- Please use our reference format:

- Please make sure that all the funding information is also entered into the online submission system and that it is complete and similar to the one in the acknowledgement section of the manuscript text file. Presently, the institutional startup funds at Duke University are missing in the submission system. Please check.

- The Supplementary Tables 1-5 are datasets. Please upload these files as separate datasets using the filename 'Dataset EVx'. Please put a legend for these on the first TAB. Moreover, please add callouts for these files to the manuscript text using the same name ('Dataset EVx').

- All Materials and Methods need to be described in the main text using our 'Structured Methods' format, which is required for all research articles. According to this format, the Methods section should include a Reagents and Tools Table (listing key reagents, experimental models, software, and relevant equipment and including their sources and relevant identifiers), uploaded as separate file, followed by a Methods section in which we encourage the authors to describe their methods using a step-by-step protocol format with bullet points, to facilitate the adoption of the methodologies across labs. More information on how to adhere to this format as well as downloadable templates (.doc) for the Reagents and Tools Table can be found in our author guidelines (section 'Structured Methods'):

- Thank you for providing the requested source data. Please upload this as one folder per main figure (with all files for one figure in one folder and ZIPed).

In addition, I would need from you uploaded separately:

- a short, two-sentence summary of the manuscript (not more than 35 words).

- two to four short (!) bullet points highlighting the key findings of your study (two lines each).

- a schematic summary figure as separate file that provides a sketch of the major findings (not a data image) in jpeg or tiff format (with the exact width of 550 pixels and a height of not more than 400 pixels) that can be used as a visual synopsis on our website.

Yours sincerely,

Arbitrator:

In this manuscript, Pierson Smela et al. identified transcription factors (TFs) that enhance human PGCLC induction and rapidly differentiate into induced oogonia-like cells (iOLCs). While their in silico approach to identify candidate genes is potentially intriguing, some aspects of the data presentation for both hPGCLCs and iOLCs remain inadequate, even after revision. In addition, I am unconvinced regarding the maintenance of iOLCs in vitro due to lack of solid supporting data. Below, I outline several concerns that need to be addressed.

1. Flow cytometry data

hPGCLC population: In response to the reviewers' comment, the authors have added data in Figure S2F showing that all NANOS3-mVenus positive cells express EpCAM/ITGA6. However, the percentage of EpCAM/ITGA6 positive cells appears higher than the NANOS3-mVenus positive population in Figure S2F, which is presented to be less than 0.5% in the control experiments shown in Figure S4. I recommend reanalyzing the data to clarify how many of the EpCAM/ITGA6 positive cells express NANOS3-mVenus both in monolayer and aggregate induction protocols. This would help to determine the proportion of

EpCAM/ITGA6 positive cells that may include non-PGCLCs in this study. In addition, the authors mention using CD38 as a germ cell marker in the methods. Providing information on the overlap of CD38 with NANOS3-mVenus/EpCAM/ITGA6 would be informative for readers.

Gating strategy: In Figures S4 and S5A, the authors should revise the gating strategy for identifying positive cell populations. The current gate appears to include autofluorescent signals visible at the tip of the negative population. The authors should redraw the gate diagonally based on negative controls or single-staining controls.

Geometric Mean Fluorescence Intensity (Figure 2B): It is unclear how the authors estimated the geometric mean fluorescence intensity. The gating strategy for the representative genes and control should be clearly shown.

2. Immunofluorescence analysis (IF) for hPGCLCs and iOLCs

hPGCLC: In Figure S6, the authors stained ITGA6 for hPGCLCs in the aggregates, but the staining for ITGA6 appears nearly ubiquitous in flow cytometry analysis in Figure S5. Additionally, some signals are detected in the cytoplasm and nucleus, although ITGA6 is a surface marker. The authors should repeat the experiment carefully, using different combinations of antibodies, such as TFAP2C, BLIMP1, and NANOG. It would also be useful to employ an anti-GFP antibody to stain NANOS3-mVenus and check the colocalization with other germ cell markers.

iOLCs: In Figure S7, the authors stained iOLCs and claimed that the iOLCs are negative for meiosis-related markers SYCP3 and RAD51. However, without appropriate positive controls for these antibodies, it is difficult to confirm that the cells are truly negative for these markers.

3. Similarity between iOLCs and human oogonia

In response to the reviewers' comments, Figures 5 and 6 seem to be improved. In addition to the representative gene expression shown in Figure 5, the authors should include PCA or UMAP analyses using more genes to better clarify the transcriptomic similarity between iOLCs and human oogonia. This would provide a more comprehensive view of how closely the induced cells resemble natural oogonia at the transcriptional level.

4. Culture of iOLCs

The authors claim that iOLCs can be maintained *in vitro* for several weeks. However, it is unclear whether iOLCs can be autonomously maintained under these conditions, as the authors are adding Dox to activate exogenous transgenes. Can iOLCs be maintained without Dox? If so, it would suggest that the iOLCs induced in the manuscript are capable of proliferating via their own gene regulatory network as oogonia.

Moreover, in Figure 4, the DDX4 expression in "Post-Sort Day 14" is very faint, almost at the same level as the background in "Pre-Sort IF," raising doubts about the *in vitro* maintenance of iOLCs. Additional robust evidence, such as flow cytometry and transcriptome analysis after culture, is needed to support this claim.

In this manuscript, Pierson Smela et al. identified transcription factors (TFs) that enhance human PGCLC induction and rapidly differentiate into induced oogonia-like cells (iOLCs). While their in silico approach to identify candidate genes is potentially intriguing, some aspects of the data presentation for both hPGCLCs and iOLCs remain inadequate, even after revision. In addition, I am unconvinced regarding the maintenance of iOLCs in vitro due to lack of solid supporting data. Below, I outline several concerns that need to be addressed.

1. Flow cytometry data

hPGCLC population: In response to the reviewers' comment, the authors have added data in Figure S2F showing that all NANOS3-mVenus positive cells express EpCAM/ITGA6. However, the percentage of EpCAM/ITGA6 positive cells appears higher than the NANOS3-mVenus positive population in Figure S2F, which is presented to be less than 0.5% in the control experiments shown in Figure S4. I recommend reanalyzing the data to clarify how many of the EpCAM/ITGA6 positive cells express NANOS3-mVenus both in monolayer and aggregate induction protocols. This would help to determine the proportion of EpCAM/ITGA6 positive cells that may include non-PGCLCs in this study. In addition, the authors mention using CD38 as a germ cell marker in the methods. Providing information on the overlap of CD38 with NANOS3-mVenus/EpCAM/ITGA6 would be informative for readers.

We thank the referee for these important considerations of the underlying data. The data of Figure S2F (Now Appendix Figure 2) is utilized to verify that the NANOS3 reporter is functional and overlaps with a known hPGCLC population. Therefore, we utilized a well described and validated floating aggregate method which is known to induce higher efficiency hPGCLC formation (identified as EpCam/ITGA6 dual positive cells) compared to the monolayer method used for later screening experiments in Figure S4A (now Figure EV1). We likewise use this floating aggregate method for validation of TF-induced outcomes in Figure S5A (Now Figure EV2). We thank the referee for identifying this discrepancy and we have corrected the description in the figure legend, which incorrectly stated it was generated using the monolayer (lower efficiency) method.

In the manuscript, the NANOS3 reporter was used as a screening tool to prioritize transcription factors for further follow up and not used to definitively characterize the resulting cells as hPGCLCs. The monolayer screen, whose results are seen in Figure S4 (Now Figure EV1) did not use any cell surface staining, only the NANOS3 marker, and therefore re-analysis of the data to determine cell surface marker expression is not possible. Since a single marker is not sufficient to characterize cells as hPGCLCs, in Figure S4 (Now Figure EV1) the Y-axis of the graph is labeled NANOS3-mVenus+ Cells (%) and not % hPGCLCs. Using this outcome, we then identify DLX5, HHEX, and FIGLA as TFs that may be capable of improving hPGCLC yield, which we then definitively show using traditional floating aggregate differentiation methods, cell staining, gene expression, and cell surface based multi-marker flow cytometry in Figure S5 (Now Figure EV2) and S6 (Figure EV3), in line with previous studies. We therefore feel that while it is of interest to know the overlap of cell surface markers with NANOS3, that within the scope and claims of this manuscript it would not change the conclusions made, as

NANOS3 is used as TF-of-interest identification tool and not a cell phenotypic characterization method.

Gating strategy: In Figures S4 and S5A, the authors should revise the gating strategy for identifying positive cell populations. The current gate appears to include autofluorescent signals visible at the tip of the negative population. The authors should redraw the gate diagonally based on negative controls or single-staining controls.

We thank the reviewer for this suggestion. However, in Figure S4 and S5 (Now Figure EV1 and EV2), the No TF condition shown is not an outright negative control used for negative gating, it is the condition that still performs differentiation but without the additional expression of any TF. Therefore a low number of positive cells are expected and these are not autofluorescent cells. The negative control (non-fluorescently marked cells or staining controls) for gating determination is not shown in these graphs and is used to draw the negative gate utilized.

Geometric Mean Fluorescence Intensity (Figure 2B): It is unclear how the authors estimated the geometric mean fluorescence intensity. The gating strategy for the representative genes and control should be clearly shown.

We thank the reviewer for this suggested revision. We have added a description of the method used to determine geometric mean fluorescent intensity to Materials and Methods in the FACS section and added a plot of the flow graphs. Briefly, upon identification of the positive cell population after gating, the fluorescent intensity of the positive population is plotted versus the percent positive cells. For graphing, the geometric mean of the fluorescent intensity of the reporter positive cells is used for plotting.

2. Immunofluorescence analysis (IF) for hPGCLCs and iOLCs

hPGCLC: In Figure S6, the authors stained ITGA6 for hPGCLCs in the aggregates, but the staining for ITGA6 appears nearly ubiquitous in flow cytometry analysis in Figure S5. Additionally, some signals are detected in the cytoplasm and nucleus, although ITGA6 is a surface marker. The authors should repeat the experiment carefully, using different combinations of antibodies, such as TFAP2C, BLIMP1, and NANOG. It would also be useful to employ an anti-GFP antibody to stain NANOS3-mVenus and check the colocalization with other germ cell markers.

We thank the reviewer for this analysis. Figure S5 and S6 (Now Figure EV2 and EV3) use different antibodies and methods for quantifying/qualifying hPGCLCs, which likely results in different staining profiles. For the IF images, we sought to use settings that best reflect high expression of ITGA6 versus low level background expression/staining. In the Flow cytometry, we gated for the clearly separated dual positive population, as has been well described in other studies that use this differentiation method.

As flow cytometry harvests all cells in the aggregate, it is a method that can be well-utilized for quantification compared to IF which only images a single plane of the 3D aggregate. Therefore, while the absolute number of positive cells may differ slightly, the relative difference between conditions remains qualitatively similar. Our intent with Figure S5 and S6 (Now Figure EV2 and EV3) is to show that the resulting cells produced by TF-assisted differentiation show the appropriate canonical hPGCLC markers (Figure EV2A), gene expression (Figure EV2B) and protein expression (Figure EV3), which we feel is sufficient to demonstrate the cell phenotype.

iOLCs: In Figure S7, the authors stained iOLCs and claimed that the iOLCs are negative for meiosis-related markers SYCP3 and RAD51. However, without appropriate positive controls for these antibodies, it is difficult to confirm that the cells are truly negative for these markers.

We thank the reviewer for this suggested addition. We have added positive controls to the revised manuscript to validate the staining works as intended and the cells are in fact negative. These images were taken in the same experiment, but we initially omitted them from this manuscript because we were preparing a separate manuscript focused on meiosis induction.

3. Similarity between iOLCs and human oogonia

In response to the reviewers' comments, Figures 5 and 6 seem to be improved. In addition to the representative gene expression shown in Figure 5, the authors should include PCA or UMAP analyses using more genes to better clarify the transcriptomic similarity between iOLCs and human oogonia. This would provide a more comprehensive view of how closely the induced cells resemble natural oogonia at the transcriptional level.

We thank the reviewer for noting this suggestion. We have performed a UMAP between iOLC samples and human oogonia samples from the several studies listed in Figure 5. As can be seen in the revised manuscript in Appendix Figure 4B, our iOLC samples cluster very strongly with total Fetal Ovary samples (Yatsenko, et al. 2019) and hiPSC-differentiated putative human oogonia from Yamashiro, et al. 2018. iOLCs did not cluster with putative human oocytes that were sequenced in the Zhang, et al. 2017 paper, as expected as can be seen in Appendix Figure 4C. We provide the normalized RNA-seq matrix and UMAP code for reproducibility.

4. Culture of iOLCs

The authors claim that iOLCs can be maintained in vitro for several weeks. However, it is unclear whether iOLCs can be autonomously maintained under these conditions, as the authors are adding Dox to activate exogenous transgenes. Can iOLCs be maintained without Dox? If so, it would suggest that the iOLCs induced in the manuscript are capable of proliferating via their own gene regulatory network as oogonia.

Moreover, in Figure 4, the DDX4 expression in "Post-Sort Day 14" is very faint, almost at the same level as the background in "Pre-Sort IF," raising doubts about the in vitro maintenance of

iOLCs. Additional robust evidence, such as flow cytometry and transcriptome analysis after culture, is needed to support this claim.

We agree that the term maintenance, given the data presented, may be a mis-representative statement in the context of genetic maintenance of the cells' phenotype. We have therefore revised the description to 'Culture and expansion' as the cells remain alive in culture and increase in number as is seen in Figure S7 (Now Figure EV4). While the post-sort Day 14 is faint, it is reproducibly shown with a different antibody in Figure S7C (Now Figure EV4) as well as using the TdTomato reporter in Figure 4A. We therefore believe that DDX4 expression continues in at least a subset of the cells. However, we have softened the claim to 'culture and expand' showing the cells can remain alive after sorting and continue to divide and expand in number and have added more context in the discussion and limitations.

Dear Dr. Kramme,

Thank you for the submission of your further revised manuscript to our editorial offices. I have already forwarded to you the report from the arbitrator that I asked to re-evaluate your study, you will find again below. As you know, the arbitrator has remaining concerns and suggestions to improve the study. Going through your further revision plan (preliminary p-b-p-response), I think the remaining points of the referee will be adequately addressed.

I thus invite you to revise the manuscript further as indicated in your letter. Please also provide a final p-b-p-response addressing the remaining points.

Moreover, I have these final editorial requests:

- Please use our reference format:

- Please provide a schematic summary figure as separate file that provides a sketch of the major findings (not a data image) in jpeg or tiff format (with the exact width of 550 pixels and a height of not more than 400 pixels) that can be used as a visual synopsis on our website. The provided image has the wrong size and scaling it down results too small fonts.

- Please separate the single images in figure 4B using white lines.

- Please make sure the final Appendix file is provided as pdf and with the items named as 'Appendix Figure Sx'.

Best,

Achim Breiling
Achim Editor
EMBO Reports

Arbitrator:

Unfortunately, the authors have not addressed some of my concerns regarding the representation of flow cytometry data. Furthermore, the additional data presented to demonstrate the similarity between iOLCs and human oogonia raise another concern, as outlined below.

1. Flow cytometry data:

The authors still state in the legend of Appendix Figure S2F: "flow cytometry analysis of hPGCLC formation using the method

used for screening (Figure 2A) for the D4TN3V....." Does this indicate that the data were obtained using the monolayer method rather than the floating aggregate method? If not, the legend should be corrected for accuracy. Additionally, I suggest including the exact percentages of EpCAM/ITGA double-positive cells and NANOS3-mVenus-positive cells in Appendix Figure S2F to provide a clearer quantitative representation.

2. Gating strategy:

In Figure EV2B, the absence of appropriate non-staining or single-staining controls in the paper makes it unclear how the authors defined their gating strategy. Notably, the cells at the tip of the negative cell population, which are included as part of the double-positive population, clearly represent a different population from the truly positive cells. Including these cells could significantly affect the calculated percentage of positive cells. Furthermore, I am curious why the staining pattern in Figure EV2 differs so markedly from that in Appendix Figure S2F, where the separation of populations appears much clearer.

3. Similarity between iOLCs and human oogonia:

In Appendix Figure 4B, while D3 iOLCs appear similar to xrOvary and fetal ovary samples, D5 forms distinct clusters. In the revised manuscript, the authors focus exclusively on D3, and Appendix Figure 4C includes only D3 in the analysis. Does this suggest that D5 does not meet the criteria for an oocyte-like phenotype, despite being treated similarly to D3 in the preceding section? If the authors assert that D3 and D5 are comparable, they should clarify why Appendix Figure 4B shows distinct clustering for D5. Additionally, D5 should be included in the dataset used for Appendix Figure 4C to ensure a comprehensive and unbiased analysis.

Reviewer Requests:**Point by Point:**

The authors still state in the legend of Appendix Figure S2F: "flow cytometry analysis of hPGCLC formation using the method used for screening (Figure 2A) for the D4TN3V....." Does this indicate that the data were obtained using the monolayer method rather than the floating aggregate method? If not, the legend should be corrected for accuracy. Additionally, I suggest including the exact percentages of EpCAM/ITGA double-positive cells and NANOS3-mVenus-positive cells in Appendix Figure S2F to provide a clearer quantitative representation.

Thank you for pointing this discrepancy out, as part of the revision we corrected an error in the description of Appendix Figure S2F but did not update the legend. This figure is generated using the floating aggregate method, this has been corrected in the revised submission. We have included the exact percentages in the revised Figure S2F for a clearer representation. Specific changes in the revised manuscript:

In the results section, we clarified how the reporters were validated for Appendix Figure 2E and 2F: "Using flow cytometry after CRISPR-activation induction of the DDX4 reporter and floating aggregate hPGCLC induction, we confirmed that the reporter lines induce expression of the fluorescent markers upon induction of the target genes, and for the

NANOS3 reporter, overlaps with known hPGCLC markers EpCAM/ITGA6 (Appendix Figure S2E, S2F)."

In the legend of Appendix Figure S2F: "Flow cytometry analysis of hPGCLC formation using the floating aggregate method (Sasaki et al. 2015) utilized in Figure EV2 for the D4TN3V reporter line. Live, single cells are visualized for expression of the cell surface markers EpCAM and ITGA6 (left panel). Additionally in the subset of NANOS3-mVenus+ cells, expression of EpCAM/ITGA6 is visualized (right panel)."

In Appendix Figure S2F, we have added the exact percentage of double-positive cells and NANOS3-mVenus-positive cells: Double Positive out of total live ~15%, Double Positive out of NANOS3-mVenus-positive cells: 87%.

We believe these updates and clarifications sufficiently provide the clarity requested by the reviewer.

In Figure EV2B, the absence of appropriate non-staining or single-staining controls in the paper makes it unclear how the authors defined their gating strategy. Notably, the cells at the tip of the negative cell population, which are included as part of the double-positive population, clearly represent a different population from the truly positive cells. Including these cells could significantly affect the calculated percentage of positive cells. Furthermore, I am curious why the staining pattern in Figure EV2 differs so markedly from that in Appendix Figure S2F, where the separation of populations appears much clearer.

Figure EV2B is a gene expression graph. We believe the arbitrator is referencing Figure EV2A lower panel. In the revised submission, Figure EV2A lower panel now includes the non-staining and single staining reference controls to more clearly define how the gating was performed. Additionally, we have revised the gating to only include the distinct population of dual positive cells and have updated the corresponding graph of Figure EV2A (upper panel). As can be seen, this does not change the interpretation of results as the inclusion of doxycycline for the DLX5, HHEX, and FIGLA induction conditions significantly improves the dual positive hPGCLC yield in the updated dual-positive gated population.

The population is not as clearly defined in Appendix S2F, likely due to minor differences in staining and flow cytometry. However, Appendix Figure 2F is solely utilized to validate reporter and identity expression whereas Figure EV2A is used as a quantitative comparison with appropriate replicates. In Figure EV2A, the separation is indeed clearer which enables and supports a more precise quantitative analysis that is used in the interpretation of results.

We believe these updates and clarifications sufficiently address the concerns of the reviewer.

In Appendix Figure 4B, while D3 iOLCs appear similar to xrOvary and fetal ovary samples, D5 forms distinct clusters. In the revised manuscript, the authors focus exclusively on D3, and Appendix Figure 4C includes only D3 in the analysis. Does this suggest that D5 does not meet the criteria for an oocyte-like phenotype, despite being treated similarly to D3 in the preceding section? If the authors assert that D3 and D5 are comparable, they should clarify why Appendix Figure 4B shows distinct clustering for D5. Additionally, D5 should be included in the dataset used for Appendix Figure 4C to ensure a comprehensive and unbiased analysis.

Thank you for the analysis of the results. To provide a more consistent and comprehensive analysis, we have streamlined Appendix Figure S4. Appendix Figure S4A provides a cell type classification analysis of reference samples, iOLCs, and external controls. In the updated manuscript, we removed the old Appendix Figure S4B and S4C and replaced them with a single UMAP figure (now Appendix Figure S4B) that includes all samples listed in Appendix Figure S4A. We believe this ensures consistency between the two figure panels and gives a more robust, global assessment of similarity.

As can be seen, in this more unbiased and comprehensive analysis, the D3 and D5 samples treated with DNMTi readily cluster with the Yatsenko Week 19 in vivo oogonia. Interestingly, the D3 samples without DNMTi cluster more readily with recently described early-oogonia methods (Murase et al. 2024). We do see that Yamashiro et al. 2019 xrOvary samples form a distinct cluster from both the samples in this manuscript as well as the samples from the same authors using a 2D expansion method (Murase et al. 2024).

To address the reviewer's concerns around differences between D3 and D5 (and broadly different TF combinations that yield iOLCs) we have included Dataset EV4 and EV5 and discussion of differential gene expression and GO term enrichment between conditions. In the results section this is detailed saying:

“Next, we performed a differential gene expression analysis to identify differences among iOLCs induced using different TFs. In comparison to the D3 TF combinations, the other TF combinations showed significant changes to gene expression (Dataset EV4). We conducted a GO term enrichment analysis on differentially expressed genes. AR-V7 overexpression did not result in activation of spermatogenesis-related genes; the only enriched GO term among upregulated genes was “chromatin organization”, and contrary to our expectations, “male sex differentiation” was enriched among downregulated genes. We found that overexpression of ANHX (relative to D3 control) upregulated genes related to reproductive development (including “spermatogenesis”, “sexual reproduction”, “gamete generation”, “meiotic cell cycle”, “developmental process involved in reproduction” and “retrotransposon silencing”), and downregulated genes related to mesenchymal and cardiac development. For overexpression of ZGLP1, no GO terms were significantly enriched in the differentially expressed genes. However, the D5 combination (overexpression of both ANHX and ZGLP1) showed more extensive changes. Upregulated genes were enriched for GO terms related to metabolism, meiosis, germ cell development, and oogenesis (Dataset EV5). GO terms enriched among

downregulated genes were mainly related to cell adhesion, migration, blood vessel development, and angiogenesis. These results indicate that the two additional factors in the D5 combination act to reinforce germ cell development and downregulate off-target differentiation.”

Furthermore, we have added a new Appendix Figure S5 that focuses on the difference in gene expression in the presence and absence of DNMTi. Comparison of D3 without DNMTi compared to all samples shows a difference in gene expression in which GO term enrichment shows important pathways related to development, meiosis, recombination, and cell cycle.

Based on these findings and the totality of data, we conclude that both D3 and D5 generated DDX4+ iOLCs that share strong transcriptomic and marker expression similarity to oogonia. Further, addition of DNMTi improves not only the yield of oogonia, but also the gene expression profile, epigenetic profile, and related developmental progression towards meiosis, seen in the GO term enrichment, similarity to in vivo week 19 oogonia transcriptomically, and similarity to Week 5.5 oogonia epigenetically.

We have added to the results section the following referencing the new Appendix Figure S4B and Appendix Figure S5:

“Furthermore, UMAP analysis demonstrated that D3 and D5-derived iOLCs from DNMTi treatment were readily clustered with Week 19 Fetal oogonia (Appendix Figure S4B). Interestingly, the D3-derived iOLCs without DNMTi clustered more readily with recently described in vitro oogonia samples from Murase et al. 2024 (Appendix Figure S4B). DGE and GO term enrichment demonstrated that DNMTi significantly enriched gene expression related to meiosis, cell cycle, recombination, and development (Appendix Figure S5).”

In the discussion section we likewise have added:

“In our optimized protocol for iOLC production, we found that inhibition of DNMT1 efficiently erases DNA methylation and leads to an epigenetic state closely resembling that of human fetal germ cells. Additionally, we find that DNMTi promotes expression of meiosis and development related gene pathways and results in iOLCs with more transcriptomic resemblance to in vivo oogonia (Appendix Figure S4 and S5).”

We believe these changes address the concerns of the reviewer and provide a more comprehensive and unbiased interpretation of the transcriptomic data and clarification of the phenotypes obtained from different induction methods.

Dr. Christian Kramme
Wyss Institute at Harvard Medical School
United States

Dear Dr. Kramme,

Thank you for the submission of your finally revised manuscript to our editorial offices. I went through your final p-b-p-response and consider the remaining issues by the arbitrator and the editorial requests as adequately addressed.

I am thus very pleased to accept your manuscript for publication in the next available issue of EMBO reports. Thank you for your contribution to our journal.

Yours sincerely,
